# Towards Robust Multi-Agent Imitation Learning via Global Credit Sequence Decoding

## Abstract

Multi-Agent Reinforcement Learning (MARL) has emerged as a promising approach to solving complex decision-making problems such as multi-agent collaboration. To avoid the difficulty of designing complex reward functions, researchers increasingly adopt imitation learning. Classical methods extend single-agent imitation learning to multi-agent settings by matching distributions from expert demonstrations. However, noisy or low-quality trajectories within these demonstrations can mislead joint policy optimization, leading to significant performance degradation. This study introduces a sequential autoregressive architecture that models global dependencies among agents, facilitating adaptive credit assignment and policy optimization. The architecture theoretically enhances the variance of joint advantages and rewards, addressing issues like vanishing gradients and mode collapse caused by noisy demonstrations. Experiments show that our method achieves a significant performance improvement on multiple benchmarks.

## 1 Introduction

In recent years, Multi-Agent Reinforcement Learning (MARL) has made progress in many real-world applications involving multi-agent collaboration, such as dexterous robotic control Chen et al. (2022), multi-drone coordination Yun et al. (2022), and multi-player games Mnih et al. (2013); Schulman et al. (2017); Chen et al. (2021). A core challenge in reinforcement learning lies in designing explicit reward functions for complex tasks, as imperfect rewards can undermine the robustness of online learning Russell (1998); Ng & Russell (2000); Fu et al. (2017); Hadfield-Menell et al. (2017). Compared to single-agent reinforcement learning, multi-agent reinforcement learning is particularly difficult due to the interdependence of goals and the implicit and unstructured relationships between agents, making it challenging to define appropriate reward functions for each individual agent Song et al. (2018); Wang et al. (2021).

Imitation Learning (IL) Hadfield-Menell et al. (2016) offers an alternative solution by training using expert demonstrations without the need for explicit rewards. This approach treats learning as a distribution matching problem, aiming to approximate the expert policy. By formalizing the problem as a distribution matching issue for individual policies, generative adversarial frameworks have shown great potential in imitation learning Ho & Ermon (2016); Song et al. (2018). However, multi-agent demonstrations often contain noisy actions, and rigidly imitating joint actions can lead to performance degradation, as joint actions do not necessarily guarantee the optimality of each agent. For example, in multi-player soccer, a player's defensive move may contribute little to an offensive play, while other players' passing or shooting actions could be crucial, as shown in Figure (1). Simply matching the actions of all agents may mislead the optimization, weakening the model's ability to capture key collaborative patterns, thus impairing overall performance.

A key challenge in multi-agent imitation learning is how to decompose each agent's contribution to the team outcome in order to provide accurate rewards. Traditional attribution methods assume that individual contributions are independent and linearly additive, neglecting complex dependencies Yang et al. (2018); Zhan et al. (2019); Le et al. (2017); Yu et al. (2019). This limitation is especially prominent in generative adversarial frameworks, where overlooking dependencies can lead to insufficient reward variance and vanishing policy gradients Zhang et al. (2022). Independent reward generation further exacerbates learning imbalances, hindering the distinction between effective and noisy actions, and leading to unclear optimization directions for joint policies Wang et al. (2021).

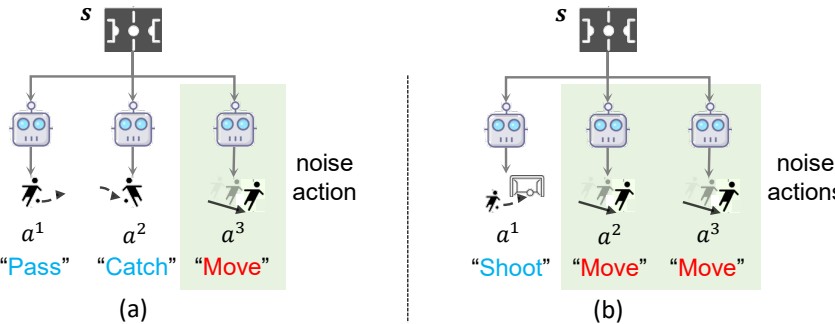

Figure 1: Taking multi-player soccer as an example, some actions in the joint policy may be non-critical and noisy. (a) Player 1 passes the ball to Player 2, while Player 3 performs a run unrelated to the pass. (b) Player 1 is executing a shooting action, while Player 2 and Player 3 are making movements unrelated to the shot. The movement actions mentioned ahead contributes little to the pass or the shot and can be regarded as noise, even though it may play other potential roles in the overall team policy.

To address these issues, it is necessary to model the dependency structure between agent actions. Existing studies have proposed modeling opponents to capture correlated policies, but this introduces additional overhead and redundancy Tian et al. (2019); Liu et al. (2020). In this paper, we propose MILD$^2$, a Transformer-based generative adversarial framework designed to capture multi-agent dependencies. This framework can approximate the joint reward distribution, magnify the reward and advantage differences during training, and alleviate convergence issues caused by the mismatch in learning speeds between the generator and discriminator. The main contributions of this paper are as follows:

- We reveal the limitations of traditional independent multi-agent imitation learning frameworks in modeling complex dependencies, which restrict reward and advantage differentiation and hinder convergence.

- We provide theoretical evidence that introducing dependency modeling for joint reward and policy distribution matching outperforms traditional methods.

- We propose a Transformer-based autoregressive generator and discriminator that implement adaptive serialization attribution and policy optimization, thereby reducing the impact of noisy actions.

- We demonstrate through experiments on three collaborative benchmark tasks that the proposed method significantly improves convergence speed, cumulative reward, and robustness compared to baselines.

## 2 BACKGROUND & PRELIMINARIES

### 2.1 MARKOV GAMES

Cooperative multi-agent reinforcement learning (MARL) problems are commonly formulated as Markov games, which extend the framework of Markov decision processes (MDPs) Littman (1994). A Markov game for $N$ agents is defined by a tuple $(N, \mathcal{S}, \{\mathcal{A}_i\}_{i=1}^N, \mathcal{T}, \{\mathcal{R}_i\}_{i=1}^N, \gamma)$, where $\mathcal{S}$ denotes the set of states and $\{\mathcal{A}_i\}_{i=1}^N$ represents $N$ sets of actions. The transition function $\mathcal{T} : \mathcal{S} \times \mathcal{A}_1 \times \ldots \times \mathcal{A}_N \to P(\mathcal{S})$ describes the state transition process, where $P(\mathcal{S})$ is the set of probability distributions over $\mathcal{S}$. Given that the system is in state $s_t$ at timestep $t$, the agents jointly take actions $(a^1, \ldots, a^N)$, and the state transitions to $s_{t+1}$ with probability $T(s_{t+1}|s_t, a_t^1, \ldots, a_t^N)$. The joint policy $\pi_\theta = [\pi_{\theta_1}, \ldots, \pi_{\theta_N}]$ represents the vector of individual agent policies. Occasionally, we may omit the policy parameters $\theta$ for convenience. It is worth noting that each agent has access to the complete state information. Let $i^{1:N}$ be a permutation of $N$ agents. To refer to a subset of agents from $i^k$ to $i^j$ ($1 \leq k \leq j \leq N$), we employ the subscript notation $k : j$, such that $\pi^{k:j}$ denotes the agent policies $\{\pi^k, \pi^{k+1}, \ldots, \pi^j\}$. Each agent $i$ is associated with an individual reward function $\mathcal{R}_i : \mathcal{S} \times \mathcal{A}_i \times \ldots \times \mathcal{A}_N \to \mathbb{R}$. The objective of each agent is to maximize its expected return, given

by $\mathbb{E}_\pi \left[ \sum_{t=0}^\infty \gamma^t r_i^t \right]$. Here, $r_i^t$ denotes the reward received by agent $i$ at timestep $t$, and the discount factor $\gamma \in [0, 1)$ determines how much future rewards are discounted. The task rewards given by environments are identical across all agents in the cooperative tasks examined in this study.

## 2.2 DISTRIBUTION MATCHING FOR IMITATION LEARNING

Imitation learning is a problem scenario in which an agent aims to replicate trajectories $\{\tau_1, \tau_2, \ldots\}$ that are demonstrated by an expert policy $\pi_E$ Schaal (1996); Hussein et al. (2017); Ho & Ermon (2016). Each trajectory $\tau$ consists of state-action pairs $\{(s_0, a_0), (s_1, a_1), \ldots\}$. Several approaches have been proposed to tackle the imitation learning problem. Behavioral cloning employs supervised learning to imitate expert demonstrations and learn the policy that maximizes the likelihood Bain & Sammut (1995); Torabi et al. (2018); Fujimoto & Gu (2021). Inverse reinforcement learning (IRL) involves recovering a reward function, which can then be used to train an expert policy using reinforcement learning Ng & Russell (2000); Hadfield-Menell et al. (2016). In the context of $\text{IRL}(\pi_E)$, the objective is to retrieve a reward function that optimizes the demonstrated trajectories by $\pi_E$. GAIL Ho & Ermon (2016) interprets the imitation learning problem as matching two occupancy measures, i.e., the distribution over states and actions encountered when exploring the environment with a policy. Formally, for a policy $\pi$, it is defined as $\rho_\pi(s, a) = \pi(a|s) \sum_{t=0}^\infty \gamma^t P(s_t = s|\pi)$. GAIL draws a connection between IRL and occupancy measure matching, showing that the former is a dual of the latter:

$$IRL_\psi(\pi_E) = \text{argmin}_{\pi \in \Pi} -H(\pi) + \psi^*(\rho_\pi - \rho_E), \qquad (1)$$

where $\psi^*(x) = \sup_y x^T y - \psi(y)$ is convex conjugate of $\psi$, which could be interpreted as a measure of similarity between the occupancy measures of expert policy and agent's policy. Wang et al. (2023) view multi-agent imitation learning as a distribution matching problem. They define the state-action visitation distribution of a joint policy $\pi = [\pi^1, \ldots, \pi^N]$ as $\rho_\pi(s, \mathbf{a}) := (1 - \gamma) \prod_{i=1}^N \pi^i(a^i|s) \sum_{t=0}^\infty \gamma^t P(s_t = s|(\pi))$. Therefore, distribution matching provides a solution to the imitation learning problem. Guan et al. (2021) demonstrate that the GAIL algorithm converges to the expert policy in the single-agent case using various policy gradient techniques Guan et al. (2021), including TRPO Schulman et al. (2015). They introduce it as the following min-max problem:

$$\min_\theta \max_\phi \mathcal{L}(\theta, \phi) \qquad (2)$$
$$\text{s.t. } \mathcal{L}(\theta, \phi) := V(\pi_E, r_\phi) - V(\pi_\theta, r_\phi) - \psi(\phi).$$

Here, $V(\pi, r) = \mathbb{E}_{s_0 \sim \rho_0} \mathbb{E}_\pi [\sum_{t=0}^\infty \gamma^t r_t]$ represents the expected return starting from an initial state according to policy $\pi$ and using reward function $r(s, a)$. In the multi-agent scenario, imitation learning becomes more complex due to the involvement of multiple expert policies $\pi_{E_0}, \ldots, \pi_{E_N}$ in generating the expert trajectories Song et al. (2018); Wang et al. (2023). Successful imitation in this setting necessitates coordinating the policies of all $N$ agents.

This paper mainly focuses on the MARL problem and multi-agent adversarial generative-based imitation learning frameworks, such as MAGAIL. In this paper, we write $-i^{k:j}$ to denote the set of all agents excluding $i^k, \ldots, i^j$. We define the multi-agent state-action value function for agents $i^{1:k}$ as $\mathbf{Q}_\theta^{1:k}(\mathbf{s}, \mathbf{a}^{1:k})$, which is the expected total reward once agents $i^{1:k}$ have taken their actions. Note that for $k = 0$, this becomes the state value function; for $k = N$, this is the usual state-action value function. As such, similar to Kuba et al. (2022), we can define the multi-agent advantage function as $\mathbf{A}_\theta^{1:k}(\mathbf{s}, \mathbf{a}^{1:m}, \mathbf{a}^{1:k}), (m \le k)$, which is the advantage of agents $i^{1:k}$, playing $\mathbf{a}^{1:k}$, given $\mathbf{a}^{1:m}$.

## 3 THEORETICAL ANALYSIS

This section will discuss the main issues of imitation learning methods based on distributed matching in multi-agent scenarios when using the independent framework. From the perspective of advantage variance, we attempt to explain the possible causes of gradient vanishing in multi-agent generative adversarial algorithms represented by MAGAIL and propose the theoretical guarantee of significantly improving performance through global dependency-enhanced discriminators.

### 3.1 THE LOW-VARIANCE PROBLEM IN MAGAIL

The low reward variance problem in MAGAIL stems from two fundamental issues: training mode collapse and inadequate credit assignment. First, mode collapse arises when the generator and

discriminator learn at mismatched rates. In classical GAIL, a faster-converging discriminator quickly "wins," leaving the generator with vanishing gradients and stalled progress (Baram et al., 2017; Zhang et al., 2022). Multi-Agent extensions such as MAGAIL under noisy demonstrations worsen this imbalance: (1) Noise in expert demonstrations can shift certain joint state–action pairs to the periphery or outside the expert distribution, resulting in a simpler classification boundary. In such cases, the discriminator can distinguish expert data from policy-generated data by capturing instability patterns in the expert demonstrations—such as abnormal state transitions or atypical actions—without relying on complex, fine-grained features. This reduces the learning burden and leads to faster convergence of the discriminator; (2) Thm. (1) in (Kuba et al., 2021) demonstrates that the variance of the multi-agent policy-gradient estimator not only exceeds its single-agent counterpart but grows linearly with agent count (Kuba et al., 2021). Together, these factors slow generator learning, further tilt the training balance, and cause the discriminator to assign nearly uniform, minimal rewards—yielding vanishingly low reward variance. Second, noisy or suboptimal actions in expert demonstrations disrupt effective credit assignment. Methods like MAGAIL assume equal contribution from each agent's action and deploy independent discriminators to reconstruct rewards. This approach creates spurious correlations and cannot distinguish critical actions from noise, resulting in persistently low variance across individual rewards.

To illustrate, consider the Multi-Agent Trust Region Policy Optimization (MATRPO) loss function (Li & He, 2020; Kuba et al., 2022): $J(\theta) = \mathbb{E}_{\mathbf{s} \sim \rho_{\pi_{old}}, \mathbf{a} \sim \pi_{old}}[\frac{\pi_\theta(\mathbf{a}|\mathbf{s})}{\pi_{old}(\mathbf{a}|\mathbf{s})} A_{\pi_{old}}(\mathbf{s}, \mathbf{a})]$. Here, the gradient of $J(\theta)$ with respect to $\theta$ is proportional to the joint advantage function $A_{\pi_{old}}(\mathbf{s}, \mathbf{a})$. When discriminators provide low rewards, the differences in joint action rewards diminish, leading the discriminator to classify all agent behaviors as non-expert, even if some agents perform well individually. This results in a low-level advantage function and a bottleneck in joint policy training.

Moreover, the inadequate credit assignment issue stems from **inadequate modeling of global dependencies during reward learning**. Discriminators designed independently for individual agent fail to reflect true contributions when they share team rewards. This makes it challenging to identify beneficial actions within teamwork, potentially misleading policy optimization and affecting GAIL convergence. Overall, these frameworks exhibit poor robustness against noisy expert demonstration actions, which can lead to issues like vanishing gradients during joint policy learning. In summary, while extending GAIL to multi-agent settings, existing approaches struggle with low reward variance, impacting the differentiation of action quality and contributing to suboptimal training dynamics. To address these issues, improving MAIL requires enhancing reward variance from the discriminator and reducing generator policy gradient variance.

## 3.2 REWARD VARIANCE ANALYSIS OF THE DISCRIMINATOR

The optimization process of RL algorithms typically depends on assessing the disparity between the rewards the environment offers and the present value function. This disparity is commonly captured by the advantage function in Actor-Critic algorithms (Konda & Tsitsiklis, 1999). In the majority of multi-agent actor-critic algorithms (Li & He, 2020; Kuba et al., 2022), the joint advantage function can be computed as $\mathbf{A}_{\pi_\theta}(\mathbf{s}, \mathbf{a}) = r(\mathbf{s}, \mathbf{a}) + \gamma \mathbf{V}_\phi(\mathbf{s}') - \mathbf{V}_\phi(\mathbf{s})$, where $s'$ is the next state, $\mathbf{V}_\phi$ is a neural network parameterized by $\phi$ to approximate the individual value function of joint policy $\pi_\theta$. The optimization objective of $\mathbf{V}_\phi$ is $\min_\phi [\mathbf{R}^{cdr}(\mathbf{s}, \mathbf{a}) - \mathbf{V}_\phi(\mathbf{s})]^2$, where $\mathbf{R}^{cdr}(\mathbf{s}, \mathbf{a})$ is the cumulative discounted rewards provided by the discriminator in MAGAIL framework.

**Theorem 1.** *Let $R_\omega^{cdr}$ be the cumulative discounted individual reward given by the discriminator parameterized by $\omega$, and let $V_\phi$ be the joint value function parameterized by $\phi$. Let $\mathcal{D}^\pi$ denote the total data collected by $\pi$. Then increasing the joint advantage variance $\sum_{k=1}^{N} \mathrm{Var}[A_{\pi_\theta}^k(\mathbf{s}, a_k)]$ of multi-agent policy gradients is equivalent to solving a bi-level joint value loss optimization problem that is related to the joint reward function and the joint value function:*

$$\max_w \sum_{k=1}^{N} \mathbb{E}_{\mathbf{s},a^k \in \mathcal{D}^{\pi^k}} [R_\omega^{cdr}(\mathbf{s}, a^k) - V_{\phi^*}(\mathbf{s})]^2, \tag{3}$$

$$\text{s. t. } \phi^*(\mathbf{s}) = \min_\phi \sum_{k=1}^{N} \mathbb{E}_{\mathbf{s},a^k \in \mathcal{D}^{\pi^k}} [R_\omega^{cdr}(\mathbf{s}, a^k) - V_\phi(\mathbf{s})]^2,$$

where $R_\omega^{cdr}(\mathbf{s}, a^k) = -\sum_{t=l}^{T} \gamma^{t-l}(\log \sigma(1 - D_\omega(\mathbf{s}_t, a_t^k)))$ *denotes the individual reward function for the agent* $i^k$, *and* $\sigma$ *denotes activation function (generally* Sigmoid*).*

For proof see Appendix (B.2). Thm. (1) shows that, to maximize joint-advantage variance, we should maximize the expected L2-norm of the gap between the discriminator's cumulative discounted rewards and the joint value function. Practically, this can be done by adding a reward-variance regularizer to the discriminator's loss. Furthermore, equipping the discriminator with credit-assignment—by replacing independent individua reward functions with globally dependency-enhanced ones—yields even higher advantage variance. The next theorem formalizes this gain over an independent-discriminator setup.

**Lemma 1.** *(Multi-agent Advantage Decomposition (Kuba et al., 2021)). Let* $i^{1:N}$ *be a permutation of* $N$ *agents. For any state* $\mathbf{s} \in \mathcal{S}$ *and joint actions* $\mathbf{a} = \mathbf{a}^{1:N} \in \mathcal{A}$, *the following equation holds for any subset of* $N$ *agents and any permutation of their labels:* $\mathbf{A}_{\pi_\theta}^{k+1,\dots,N}(\mathbf{s}, \mathbf{a}^{1:k}, \mathbf{a}^{k+1:N}) = \sum_{j=k+1}^{N} A_{\pi_\theta}^j(\mathbf{s}, \mathbf{a}^{1:j-1}, a^j)$, *where* $k = 0, \dots, N-1$.

**Theorem 2.** *Let* $A_{\pi_\theta}^k(\mathbf{s}, \mathbf{a}^{-k}, a^k)$ *be the advantage function of agent* $i^k$ *with global dependency-enhanced discriminator, and let* $\mathbf{A}_{\pi_\theta}(\mathbf{s}, \mathbf{a})$ *be the joint advantage function of all agents in independent framework, global dependency-enhanced discriminator framework has a more significant advantage variance compared to the independent framework:*

$$\sum_{k=1}^{N} \operatorname*{Var}_{\substack{\mathbf{s}, \mathbf{a}^{-k} \in D_t^{\pi^{-k}} \\ \mathbf{s}, a^k \in D_t^{\pi^k}}} [A_{\pi_\theta}^k(\mathbf{s}, \mathbf{a}^{-k}, a^k)] \geq \sum_{k=1}^{N} \operatorname*{Var}_{\mathbf{s}, a^k \in D_t^\pi} [A_{\pi_\theta}^k(\mathbf{s}, a^k)] \tag{4}$$

For proof see Appendix (B.2). Let $R_w^{cdr}(\mathbf{s}, \mathbf{a}^{-k}, a^k)$ be the cumulative discounted reward of $i^k$ given by the global dependency-enhanced discriminator parameterized by $w$. When optimal $\phi^*$ help keep the difference between $V_\pi(\mathbf{s})$ and $V_{\phi^*}(\mathbf{s})$ at a minimal level, similar to Eq. (11) in Appendix, we have

$$\max_w \sum_{t=0}^{\infty} \gamma^{2t} \frac{|\mathcal{D}_t^\pi|}{|\mathcal{D}^\pi|} \sum_{k=1}^{N} \operatorname*{Var}_{\substack{\mathbf{s}, \mathbf{a}^{-k} \in D_t^{\pi^{-k}} \\ \mathbf{s}, a^k \in D_t^{\pi^k}}} [A_{\pi_\theta}^k(\mathbf{s}, \mathbf{a}^{-k}, a^k)] \approx \max_w \operatorname*{\mathbb{E}}_{\substack{\mathbf{s}, \mathbf{a}^{-k} \in D_t^{\pi^{-k}} \\ \mathbf{s}, a^k \in D_t^{\pi^k}}} \sum_{k=1}^{N} [R_w^{cdr}(\mathbf{s}, \mathbf{a}^{-k}, a^k) - V_{\phi_k^*}(\mathbf{s})]^2. \tag{5}$$

The theorems above showcase that employing a global dependency-enhanced discriminator and generator to model joint reward/policy distribution matching, as opposed to the conventional independent architecture, facilitates enhanced reward variance and advantage variance during the model's training. This approach effectively mitigates the concern of disparate training speeds between the generator and discriminator to achieve better.

# 4 METHODOLOGY

This section introduces a **M**ulti-agent generative adversarial **I**mitation **L**earning framework via global **D**ependency-enhanced **D**istribution matching (MILD$^2$), which focuses on modeling complex agent dependency structure and collaborative actions sequentially autoregressively.

## 4.1 GLOBAL DEPENDENCY-ENHANCED DISCRIMINATOR

The reward function formula given in Eq. (5) indicates that the design of the discriminator should consider global state and action information to establish a joint reward distribution with a global perspective. It is designed to avoid misallocating credit for individual discriminatory rewards, which can mislead and delay the learning speed of the policy generator. Moreover, it addresses the challenges associated with low reward variance and policy gradient vanishing. The construction of the joint reward distribution is precisely accomplished through the discriminator, denoted as $D_w$. In evaluating the proficiency of agent $i^k$'s actions, the discriminator not only takes into account the global state $\mathbf{s}$ and the agent's action $a^k$, but also considers the actions $\mathbf{a}^{-k}$ of other agents $i^{-k}$ at the current time step. It enables the discriminator to capture the dependencies and collaborative relationships among multiple intelligent agents. The marginal distribution of the joint reward distribution corresponds to

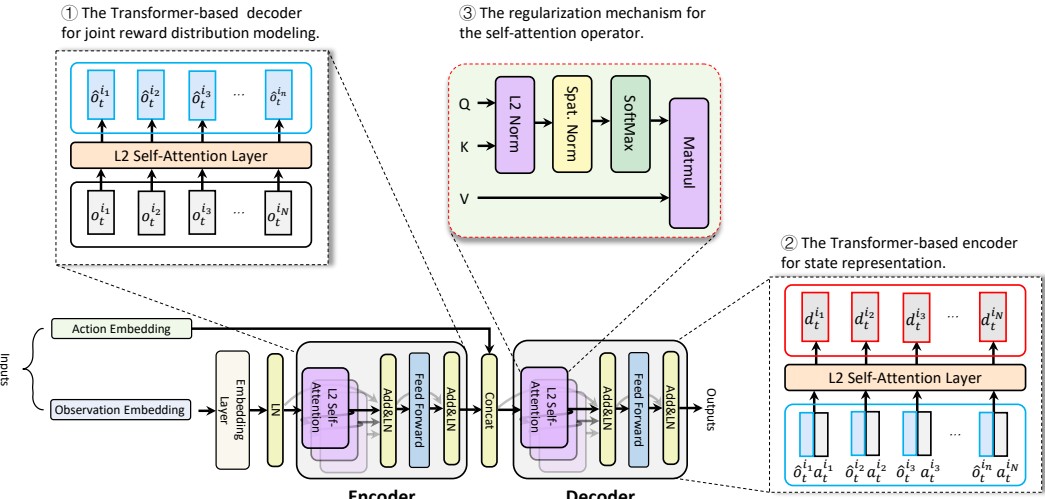

Figure 2: The Transformer-based architecture for our global dependency-enhanced discriminator. The encoder processes all agents' observations into a latent representation, which the decoder uses along with agent actions to produce expert degree scores. L2 self-attention and spectral normalization ensure Lipschitz continuity.

---

**Algorithm 1** MAIL via Global Dependency-Enhanced Distribution Matching (MILD$^2$)

---

**Input**: Initial parameters of policies $\theta_0$, discriminator $\omega_0$, and value estimator $\phi_0$; expert trajectories $\mathcal{D} = \{(\mathbf{s}, a^k)\}_{k=1}^N$; batch size $B$; Markov game as a black box $(N, \mathcal{S}, \{\mathcal{A}_i\}_{i=1}^N, \mathcal{T}, \{\mathcal{R}_i\}_{i=1}^N, \gamma)$.
**Output**: Learned policies $\pi_\theta$ and reward functions $D_\omega$.

1: **for** $u = 0, 1, 2, \ldots$ **do**
2:     Obtain trajectories of size $B$ from $\pi$ by the process: $\mathbf{s}_0 \sim \mathcal{S}, \mathbf{a}_t \sim \pi_{\theta_u}(\mathbf{a}_t|\mathbf{s}_t), \mathbf{s}_{t+1} \sim \mathcal{T}(\mathbf{s}_t|\mathbf{a}_t)$.
3:     Sample state-action pairs from $\mathcal{D}$ with batch size $B$.
4:     Denote state-action pairs from $\pi$ and $\mathcal{D}$ as $\mathcal{D}_\pi$ and $\mathcal{D}_E$.
5:     Update $\omega$ to minimize the objective Eq. (6).
6:     **for** $k = 1, \ldots, N$ **do**
7:         Compute value estimate $V^k$ and advantage estimate $A^k$ for $(\mathbf{s}, \mathbf{a}^{1:k}) \in \mathcal{S}_\pi$.
8:         Update $\phi^k$ to decrease the objective Eq. (18).
9:         Update $\theta^k$ by policy gradient with small step sizes as Eq. (19).
10:     **end for**
11: **end for**
12: **return** policy model $\pi_\theta$ and reward model $D_\omega$. =0

---

the individual reward function for each agent. A straightforward approach to meet the requirements above for discriminator design involves employing a multilayer perceptron (MLP) that takes the joint actions of the global state and agents as input. However, this design encounters two challenges: (1) It struggles to effectively capture the dynamic correlations among the actions of intelligent agents, which in turn hampers credit assignment; (2) The high dimensionality of the state and joint action inputs poses difficulties for discriminator learning. Considering the potential power of sequence modeling in the multi-agent domain (Meng et al., 2023; Wen et al., 2022), we model the global dependency structure among agents in a sequential autoregressive paradigm, and utilize the Transformer model to construct the discriminator and achieve the objectives above.

**Overall Architecture of Discriminator.** The proposed transformer-based discriminator architecture is illustrated in Fig. (2). It comprises an encoder and a decoder module. The encoder module embeds the agents' observations, constructing a global state representation. It processes a sequence of observations $\mathbf{s} = (o^1, \ldots, o^N)$ in arbitrary order through multiple blocks. Each block consists of a self-attention mechanism, an MLP layer, and residual connections to mitigate the issues of gradient vanishing and network degradation as the depth increases. The output encoding of the observations,

Table 1: Mean win rate(%, for SMAC and Football) and accumulated trajectory rewards (for Bi-dexhands and Ma-Mujoco) with standard deviation across baselines on four benchmarks.

| Benchmarks | Tasks | MAGAIL | CQL-MA | ICQ-MA | TD3-BC | OMAR | MILD$^2$(Ours.) |
|---|---|---|---|---|---|---|---|
| SMAC | 3m | 99.8±0.4 | 74.1±4.1 | 45.8±9.2 | 44.1±10.9 | **100.0±0.0** | **100.0±0.0** |
| | 3s5z | 98.0±0.7 | 6.2±2.9 | 76.0±13.6 | 17.4±12.1 | 62.8±3.7 | **100.0±0.0** |
| | 6h vs 8z | 47.3±0.2 | 0.0±0.0 | 0.3±1.4 | 0.0±0.0 | 0.0±0.0 | **96.2±0.8** |
| | MMM2 | 0.0±0.0 | 0.0±0.0 | 6.6±5.5 | 0.0±0.0 | 84.7±2.1 | **85.8±2.5** |
| Football | 3 vs 1 | 95.0±0.7 | 92.4±1.2 | 53.1±13.0 | 24.2±7.6 | 89.9±2.7 | **96.4±1.2** |
| | counterattack | 88.5±1.7 | 42.5±7.6 | 39.4±8.6 | 0.0±0.0 | 93.2±1.6 | **93.7±1.6** |
| | pass and shoot | 74.5±8.4 | 95.7±3.9 | 62.7±12.2 | 0.0±0.0 | 95.8±1.5 | **96.2±0.9** |
| Bi-DexHands | CatchOver2Underarm | 6.7±0.1 | 15.7±1.0 | 3.5±0.2 | 10.1±1.2 | 16.9±1.2 | **24.2±0.6** |
| | DoorOpenInward | 12.5±23.3 | 189.7±41.7 | -7.2±36.1 | 217.7±45.0 | 114.5±34.3 | **396.0±0.4** |
| | DoorCloseOutward | 503.3±0.1 | 839.5±11.3 | 215.37±0.1 | 41.8±5.2 | 818.8±2.4 | **1016.9±0.1** |
| Multi-Agent Mujoco | HalfCheetah 6×1 | 435.5±9.0 | 2189.5±959.4 | 3977.2±127.1 | 4123.6±146.4 | 4088.9±165.7 | **4476.0±74.8** |

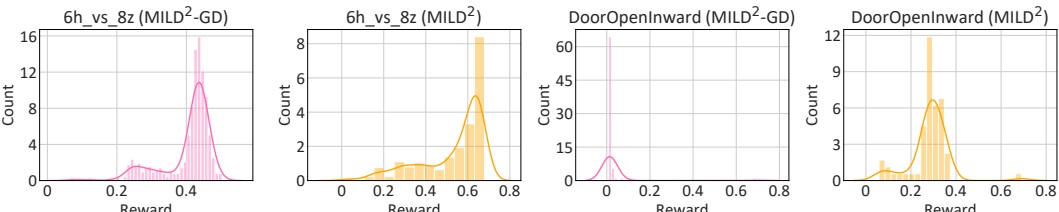

Figure 3: Distribution of rewards given by the discriminator during training. The data is collected on 6h_vs_8z (discrete actions) and DoorOpenInward (continuous actions) environments.

denoted as $(\hat{o}^1, \ldots, \hat{o}^N)$, captures the information on the agents $(i^1, \ldots, i^N)$ and the higher-level interrelationships that depict the agents' interactions. On the other hand, the decoder module evaluates the expertise level of all agents' actions and generates individual rewards accordingly. It focuses on the embedded joint action $\mathbf{a}^{1:N} = (a^1, \ldots, a^N)$ and the embedded latent state representation $(\hat{o}^1, \ldots, \hat{o}^N)$ within a sequence of decoder blocks. Crucially, each decoder block incorporates a self-attention mechanism for capturing global dependencies instead of using masked self-attention. The decoding process concludes with an MLP layer and skipping connections. The output of the final decoder block consists of a sequence of logits $(d^1, \ldots, d^N)$ representing the joint reward distribution. Moreover, the individual reward function can be calculated by $r_t^k(\mathbf{s}_t, \mathbf{a}_t^{-k}, a_t^k) = -\log \sigma(d_t^k)$, where $\sigma$ is the activation function.

**Optimization Objective of Discriminator.** We train the discriminator according to Eq. (1) and Eq. (4), where the measure of similarity $\psi = W_1^d$ for occupancy measure (distribution) matching adopts Wasserstein distance (Xiao et al., 2019). The loss function of discriminator $\mathcal{L}_d = W_d^1(\rho_\pi, \rho_E)$ is defined as:

$$\mathcal{L}_d = \sup_{r:(\mathcal{S}, \mathcal{A}) \to \mathbb{R}} \sum_{k=1}^{N} \big( \mathbb{E}_{y \sim \rho_E^{1:k}}[r(y)] - \mathbb{E}_{x \sim \rho_\pi^{1:k}}[r(x) + \underbrace{\lambda(R_\omega^{cdr}(x) - V_{\phi^*}(x))^2}_{\text{reward variance regularization}}] + \mathbb{E}_{(x,y) \sim \rho_\pi^{1:k} \times \rho_E^{1:k}}[\Omega_{d,\varepsilon}(r, x, y)]\big),$$

$$(6)$$

where $\Omega_{d,\varepsilon}(r, x, y) = -\frac{1}{4\varepsilon}(r(y) - r(x) - d(x, y))^2$ regularizes the reward function in such a way that it decreases the objective if $r_\omega(\cdot)$ is not a Lipschitz (1) function.

As shown in previous works, Lipschitz continuity is crucial for Wasserstein loss in GANs (Arjovsky et al., 2017; Xiao et al., 2019). Recent research found that transformer-based discriminators' standard dot product self-attention layer may lack Lipschitz continuity, especially in discrete action spaces. To address this, we use two regularization techniques in our discriminator (Lee et al., 2022). For more information, refer to Appendix C.1.

### 4.2 SEQUENTIAL AUTOREGRESSIVE MODELING GENERATOR

To reduce the variance of multi-agent policy gradient estimates, we also consider modeling complex dependencies among multiple agents within the generator (policy model). This mitigates convergence issues due to mismatched training speeds between the discriminator and generator. Unlike

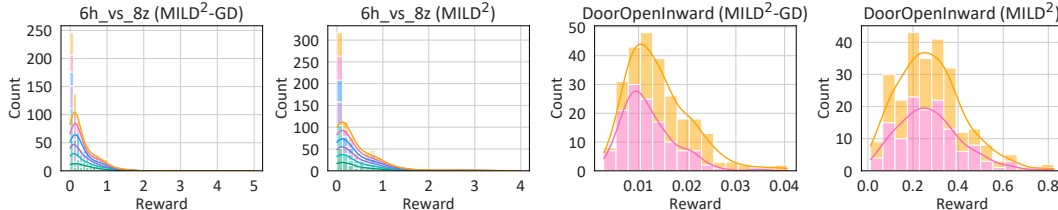

Figure 4: Statistical frequency histogram of individual rewards given by the discriminator. The data is collected on 6h_vs_8z (6 agents) and DoorOpenInward (2 agents) tasks at environmental step 1,900,000.

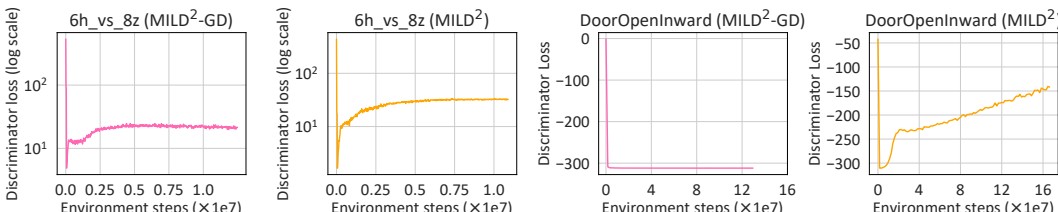

Figure 5: Discriminator loss curve comparison during training on 6h_vs_8z and DoorOpenInward environments.

classical multi-agent GAIL methods that learn independent generators for joint policy, we employ a interdependent policy generator to capture agent interactions through a sequential autoregressive model inspired by MAT (Wen et al., 2022). The proposed framework's training follows GAIL (Song et al., 2018; Wang et al., 2023). Further details are provided in Appendix (C.2).

## 5 EXPERIMENTS

The fundamental insight of MILD$^2$ revolves around a global dependency-enhanced framework for MAIL inspired by Thm. (2), as well as an encoder-decoder architecture that provides an efficient implementation for modeling and matching the joint reward and policy distribution. In this section, we evaluate the performance of MILD$^2$ on four benchmarks and compare them with state-of-the-art methods. Our experiments aim to answer two questions: (1) Does considering intricate dependencies among agents and modeling joint distributions contribute to multi-agent imitation learning models' rapid convergence and superior performance? (2) Does the discriminator architecture, fortified with global dependency, effectively augment reward and advantage variance, and alleviate the imbalance of training speed and limited robustness issues?

### 5.1 EXPERIMENTAL SETUP

**Benchmark Datasets.** We evaluated MILD$^2$ using four benchmarks: StarCraftII Multi-Agent Challenge (SMAC) benchmark (Samvelyan et al., 2019), Google Research Football benchmark (Football) (Kurach et al., 2020), Bimanual Dexterous Hands Manipulation benchmark (Bi-DexHands) (Chen et al., 2022), and Multi-Agent MuJoCo benchmark (MA-Mujoco) (de Witt et al., 2020). We constructed several multi-agent offline datasets on these benchmarks by collecting 10,000 (for tasks in Bi-DexHands) and 100,000 (for tasks in others) transitions of expert policy from HAPPO (Kuba et al., 2022).

**Baselines.** We compare our method against five classical multi-agent offline RL methods, including MAGAIL (Song et al., 2018), the multi-agent version of CQL (CQL-MA) (Kumar et al., 2020), ICQ-MA (Yang et al., 2021), TD3-BC (Fujimoto & Gu, 2021), and OMAR (Pan et al., 2022). Following most baseline methods, each algorithm runs with five seeds, where the performance is evaluated 20 times every 50 episodes. We show experimental details in Appendix (D.1).

## 5.2 Main Results

Tab. (1) shows that MILD$^2$ outperforms MAGAIL (baseline for independent distribution matching) and other advanced multi-agent imitation learning methods across four benchmarks. In discrete action space benchmarks (SMAC and Football), MILD$^2$ consistently improves accumulated trajectory rewards by 0.15% to 81.27% across various difficulty settings. It also achieves a significant 33.33% to 300% win rate improvement on challenging SMAC tasks involving complex cooperation. In continuous action space benchmarks (Bi-DexHands and MA-Mujoco), MILD$^2$ exhibits an 8.54% to 81.91% performance enhancement. Fig. (9) visually demonstrates that our proposed framework achieves faster joint distribution matching of multi-agent policies while maintaining stability, even outperforming expert policies. Introducing global dependencies through the discriminator and generator, modeling joint reward and policy distribution, enables each agent to receive more plausible individual rewards and advantages. It leads to discovering higher-rewarding behaviors beyond those demonstrated in expert demonstrations. Overall, MILD$^2$ significantly improves performance, especially in complex and challenging environments.

## 5.3 Ablation Studies

To demonstrate the effectiveness of MILD$^2$, we performed ablation experiments to showcase the enhanced effects of capturing global dependencies in distribution matching-based MAIL framework.

**Analysis on Reward Distribution.** We compared the reward distributions of MILD$^2$ and a variant called **MILD$^2$-GD**, which uses independent individual distribution matching and lacks global dependency modeling. Fig. (3) shows the discriminator's reward distributions during training for both algorithms. Fig. (3) and Fig. (4) illustrate that introducing global dependency modeling to MILD$^2$ increases reward variance and raises the median reward. It supports the effectiveness of MILD$^2$ in addressing the low reward variance issue, as hypothesized in our study.

**Analysis on Training Loss.** Moreover, MILD$^2$ effectively counteracted the pattern of discriminator loss (D-loss), initially decreasing and subsequently reaching a plateau. Fig. (5) presents a representative dataset depicting the training progression of D-loss. All parameters except the algorithms employed remained consistent between the two training. Significantly, MILD$^2$ successfully elevated the D-loss, which indicates that incorporating global dependencies through modeling facilitated a reduction in the problem of gradient vanishing after amplifying the variance of rewards. See Appendix (D.2) for detailed ablation experiments such as different model architectures.

## 5.4 Robustness Studies

We also performed robustness studies on sequential autoregressive policy and reward models, evaluating their performance in noisy environments. Empirical evaluation using a mixed dataset combining expert and medium-score policy data revealed the model's resilience across varying noise ratios. Tab. (6) shows that unlike the baseline models MAGAIL and ICQ-MA, which experienced a significant performance degradation at 50% noise levels, our model maintained a consistent 90% win rate in the 6h vs. 8z scenario. The model's ability to capture global dependencies also provides notable advantages in in-sample generalization on limited or less diverse datasets. See Appendix (D.2) for detailed robustness experiments.

## 6 Conclusion

In conclusion, this paper addresses the limitations of traditional independent multi-agent imitation learning frameworks in capturing complex dependencies among multiple agents. We highlight the importance of sequential modeling interdependencies and cooperative relationships among agents to enhance reward and advantage variances, and to facilitate model convergence. The paper proposes a Transformer-based framework for multi-agent imitation learning, leveraging the sequential modeling capabilities of the Transformer model to approximate the global distribution of reward function and policy accurately. Experimental results on cooperative benchmarks demonstrate the effectiveness of the proposed method, outperforming baseline algorithms in terms of convergence, stability, and robust adaptability. More other explicit agent dependency structures could explored in future work.

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
