

Figure 6: In a collaborative task involving two agents, labeled as $i^1$ and $i^2$, with action space $\{x, y\}$, $i^2$'s policy is strongly influenced by $i^1$'s actions, leading to varying occupancy measures for $i^2$. (a) Policy-Independent Distribution Matching: When the agents act independently, imitation learning approximates their policy occupancy measures separately. When $i^2$'s policy varies with $i^1$'s actions, distribution matching attempts to align a new distribution with both individual distributions, potentially causing more significant errors and uniform distributions. (b) Policy-Dependent (Joint) Distribution Matching: When agents are interdependent, imitation learning considers $i^1$'s actions when approximating $i^2$'s distribution. It results in joint distributions linked to other agents' actions, reducing errors and identifying advantageous actions.

## A    RELATED WORKS

### A.1    IMITATION LEARNING AND MULTI-AGENT IMITATION LEARNING

Imitation learning (IL) derives expert policies from demonstrations without requiring a reward function. IL methods are primarily categorized into Behavior Cloning (BC) and Inverse Reinforcement Learning (IRL) (Ng & Russell, 2000; Abbeel & Ng, 2004). BC maximizes the likelihood of state-action pairs in demonstrations to mimic expert behavior but often suffers from covariate shift due to its disregard for policy distribution within the environment (Ross & Bagnell, 2010; Ross et al., 2011). IRL, in contrast, recovers a reward function and optimizes the policy using reinforcement learning instead of BC's supervised learning approach. Recent adversarial methods have improved data efficiency over traditional IRL (Ho & Ermon, 2016; Finn et al., 2016; Fu et al., 2017).

In multi-agent scenarios, early approaches extended single-agent IL methods, such as TD3-BC Fujimoto & Gu (2021) for behavior cloning and MAGAIL Song et al. (2018) for inverse reinforcement learning. These methods decompose multi-agent expert demonstrations into independent single-agent trajectories, prompting agents to imitate paths independently without considering interactions or cooperative benefits. While effective for simple tasks, this approach fails in complex, large-scale cooperation. Recent research addresses this by modeling interactions among agents. For instance, Le et al. (2017) introduced a latent coordination model for cooperative games with distinct roles, reflecting scenarios like autonomous driving where agents require role-based cooperation.

Other works have explored generative adversarial imitation learning for multi-agent tasks. For example, Bhattacharyya et al. (2018) used parameter sharing but did not model agent interactions. Interaction Network (Battaglia et al., 2016) simulates physical object relationships via binary links, while CommNet (Sukhbaatar et al., 2016) learns dynamic agent communication without explicitly modeling action dependencies. Graph Neural Networks (GNNs) (Kipf et al., 2018) and attention mechanisms (Hoshen, 2017) have been proposed to infer multi-agent relationships. Additionally, Li et al. (2020) combined generative models and attention mechanisms to capture multi-agent behavior generation processes. These studies focus on reasoning about relationships rather than explicitly modeling action interdependencies.

Despite progress, most work emphasizes agent behavior prediction while offering limited insight into action interdependencies and credit assignment in reward reconstruction.

### A.2    DISTRIBUTION MATCHING IN MULTI-AGENT IMITATION LEARNING

Ho & Ermon (2016) introduced adversarial distribution matching for single-agent imitation learning through the GAIL algorithm. Building on this, Song et al. (2018) extended GAIL to multi-agent environments, framing independent imitation learning as a Nash equilibrium search under the assumption of a unique equilibrium. MAGAIL uses recent advancements in GAIL convergence

theory (Guan et al., 2021) to describe convergence towards the joint expert policy. Similarly, Wang et al. (2021) applied copula functions in multi-agent reinforcement learning (MARL) to model interdependencies between agents' marginal policies.

Other studies, such as Durugkar et al. (2020) and Radke et al. (2022), have shown the benefits of balancing individual preferences (e.g., aligning state-action visitation distributions) with shared task rewards to improve performance. Wang et al. (2023) approached multi-agent imitation learning as a decentralized distribution matching problem, combining distribution-matching rewards with task rewards.

This paper shifts the focus from imitation learning to enhancing cooperative task performance using distribution matching techniques that explicitly account for inter-agent dependencies. As shown in Fig. (6), prior work typically treats multi-agent imitation learning as a decentralized problem of independent distribution matching (Song et al., 2018). This approach often overlooks inter-agent dependencies and cooperative dynamics, leading to challenges in convergence and robustness within generative adversarial frameworks (Wang et al., 2021). In scenarios with noisy expert demonstrations or large-scale collaboration, decentralized distribution matching struggle to identify advantageous actions accurately, risking the imitation of suboptimal behaviors and hindering convergence. In contrast, our method adopts a centralized distribution matching framework that incorporates inter-agent dependencies. This approach improves the identification of advantageous actions, assigns appropriate imitation rewards, and mitigates convergence and robustness issues in complex tasks.

## B    PROOFS OF THE THEORETICAL RESULTS

### B.1    LEMMAS

**Lemma 1.** *(Variance Bound in MAPG (Kuba et al., 2021)). The CTDE (Centralized Training and Decentralized Execution) and DT (Decentralized Training) estimators of Multi-Agent Policy Gradient (MAPG) satisfy*

$$Var_{s_{0:\infty}\sim d_\theta^{0:\infty}, \mathbf{a}_{0:\infty}\sim\pi_\theta}\left[\mathbf{g}_C^i\right] - Var_{s_{0:\infty}\sim d_\theta^{0:\infty}, \mathbf{a}_{0:\infty}\sim\pi_\theta}\left[\mathbf{g}_D^i\right]$$

$$\leq \frac{B_i^2}{1-\gamma^2}\sum_{j\neq i}\epsilon_j^2 \leq (n-1)\frac{(\epsilon B_i)^2}{1-\gamma^2} \tag{7}$$

*where*

$$B_i = \sup_{s,\mathbf{a}}\left\|\nabla_{\theta^i}\log\pi_\theta^i(a^i|s)\right\|,$$

$$\epsilon_i = \sup_{s,\mathbf{a}^{-i},a^i}\left|A_\theta^i(s,\mathbf{a}^{-i},a^i)\right|,$$

*and*

$$\epsilon = \max_i \epsilon_i.$$

Lemma. (**??**) shows that the variance between MAPG and PG estimation is non-negative and can grow linearly with the number of agents Kuba et al. (2021).

**Lemma 2.** *(Multi-agent Advantage Decomposition (Kuba et al., 2021)). Let $i^{1:N}$ be a permutation of $N$ agents. For any state $\mathbf{s} \in \mathcal{S}$ and joint actions $\mathbf{a} = \mathbf{a}^{1:N} \in \mathcal{A}$, the following equation holds for any subset of $N$ agents and any permutation of their labels: $\mathbf{A}_{\pi_\theta}^{k+1:N}(s, \mathbf{a}^{1:k}, \mathbf{a}^{k+1:N}) = \sum_{j=k+1}^{N} A_{\pi_\theta}^j(\mathbf{s}, \mathbf{a}^{1:j-1}, a^j)$, where $k = 0, \ldots, N-1$.*

Lemma. (1) states that the joint advantage of agents' actions is the sum of the individual agents' multi-agent advantages. It suggests that a MARL problem can be unbiasedly decomposed into $n$ single-agent RL problems.

### B.2    PROOFS OF THEOREMS 1 AND 2

**Theorem 1.** *Let $R_\omega^{cdr}$ be the cumulative discounted individual reward given by the discriminator parameterized by $w$, and let $V_\phi$ be the joint value function parameterized by $\phi$. Let $\mathcal{D}^\pi$ denote the*

*total data collected by $\pi$. Then increasing the joint advantage variance $\sum_{k=1}^{N} \mathrm{Var}[A_{\pi_\theta}^k(\mathbf{s}, a^k)]$ of multi-agent policy gradients is equivalent to solving a bi-level joint value loss optimization problem that is related to the joint reward function and the joint value function:*

$$\max_w \sum_{k=1}^{N} \mathbb{E}_{\mathbf{s},a^k \in \mathcal{D}^{\pi^k}} [R_\omega^{cdr}(\mathbf{s}, a^k) - V_{\phi^*}(\mathbf{s})]^2, \tag{8}$$

$$\text{s. t. } \phi^*(\mathbf{s}) = \min_\phi \sum_{k=1}^{N} \mathbb{E}_{\mathbf{s},a^k \in \mathcal{D}^{\pi_k}} [R_\omega^{cdr}(\mathbf{s}, a^k) - V_\phi(\mathbf{s})]^2,$$

*where $R_\omega^{cdr}(\mathbf{s}, a^k) = -\sum_{t=l}^{T} \gamma^{t-l}(\log \sigma(1 - D_\omega(\mathbf{s}_t, a_t^k)))$ denotes the individual reward function for the agent $i^k$, and $\sigma$ be activation function (generally Sigmoid).*

*Proof.* To prove it, we need to establish the connection between the sum of individual value loss $\sum_{k=1}^{N} \mathbb{E}_{\mathbf{s},a^k \in \mathcal{D}^{\pi^k}} [R_\omega^{cdr}(\mathbf{s}, a^k) - V_{\phi^*}(\mathbf{s})]^2$ and the individual advantage function $A_{\pi_\theta}^k(\mathbf{s}, a^k)$. Let $\mathcal{D}_t^\pi$ denote the data collected starting at time step $t$. Following (Zhang et al., 2022), we have:

$$\sum_{k=1}^{N} \mathbb{E}_{\mathbf{s},a^k \in \mathcal{D}^{\pi^k}} [R_\omega^{cdr}(\mathbf{s}, a^k) - V_\phi(\mathbf{s})]^2$$

$$= N \mathbb{E}_{\mathbf{s},a^k \in \mathcal{D}^{\pi_k}} (V_\pi(\mathbf{s}) - V_\phi(\mathbf{s}))^2$$

$$+ \sum_{t=0}^{\infty} \gamma^{2t} \frac{|\mathcal{D}_t^\pi|}{|\mathcal{D}^\pi|} \sum_{k=1}^{N} \mathrm{Var}_{\mathbf{s},a^k \in \mathcal{D}_t^{\pi^k}} [A_{\pi_\theta}^k(\mathbf{s}, a^k)]. \tag{9}$$

Then we look for $\phi^*$ to minimize the loss of joint value. Ideally, $\phi^*$ would keep the difference between $V_\pi(\mathbf{s})$ and $V_{\phi^*}(\mathbf{s})$ at a small level, then we have:

$$\min_\phi \sum_{k=1}^{N} \mathbb{E}_{\mathbf{s},a^k \in \mathcal{D}^{\pi^k}} [R_\omega^{cdr}(\mathbf{s}, a^k) - V_\phi(\mathbf{s})]^2$$

$$= \min_\phi \sum_{k=1}^{N} \mathbb{E}_{\mathbf{s},a^k \in \mathcal{D}^{\pi^k}} [R_\omega^{cdr}(\mathbf{s}, a^k) - V_{\phi^*}(\mathbf{s})]^2$$

$$= N \mathbb{E}_{\mathbf{s},a^k \in \mathcal{D}^{\pi^k}} (V_\pi(\mathbf{s}) - V_\phi(\mathbf{s}))^2$$

$$+ \sum_{t=0}^{\infty} \gamma^{2t} \frac{|\mathcal{D}_t^\pi|}{|\mathcal{D}^\pi|} \sum_{k=1}^{N} \mathrm{Var}_{\mathbf{s},a \in \mathcal{D}_t^{\pi^k}} [A_{\pi_\theta}^k(\mathbf{s}, a^k)]$$

$$\approx \sum_{t=0}^{\infty} \gamma^{2t} \frac{|\mathcal{D}_t^\pi|}{|\mathcal{D}^\pi|} \sum_{k=1}^{N} \mathrm{Var}_{\mathbf{s},a \in \mathcal{D}_t^{\pi^k}} [A_{\pi_\theta}^k(\mathbf{s}, a^k)]. \tag{10}$$

Then in order to increase the joint advantage variance, we should add the maximization operator on both sides of the equation:

$$\max_w \sum_{k=1}^{N} \mathbb{E}_{\mathbf{s},a^k \in \mathcal{D}^{\pi^k}} [R_\omega^{cdr}(\mathbf{s}, a^k) - V_{\phi^*}(\mathbf{s})]^2$$

$$\approx \max_w \sum_{t=0}^{\infty} \gamma^{2t} \frac{|\mathcal{D}_t^\pi|}{|\mathcal{D}^\pi|} \sum_{k=1}^{N} \mathrm{Var}_{\mathbf{s},a^k \in \mathcal{D}_t^{\pi^k}} [A_{\pi_\theta}^k(\mathbf{s}, a^k)]. \tag{11}$$

So the original proposition that increasing the joint advantage variance is equivalent to solving a bi-level joint value loss optimization problem is proved. $\square$

**Theorem 2.** *Let $A_{\pi_\theta}^k(\mathbf{s}, \mathbf{a}^{-k}, a^k)$ be the advantage function of agent $i_k$ with global dependency-enhanced discriminator, and let $\mathbf{A}_{\pi_\theta}(\mathbf{s}, \mathbf{a})$ be the joint advantage function of all agents in independent*

*framework, global dependency-enhanced discriminator framework has a more significant advantage variance compared to the independent framework:*

$$\sum_{k=1}^{N} \operatorname*{Var}_{\substack{\mathbf{s},\mathbf{a}^{-k}\in\mathcal{D}_t^{\pi^{-k}} \\ \mathbf{s},a^k\in\mathcal{D}_t^{\pi^k}}} [A_{\pi_\theta}^k(\mathbf{s},\mathbf{a}^{-k},a^k)] \geq \sum_{k=1}^{N} \operatorname*{Var}_{\mathbf{s},a^k\in\mathcal{D}_t^{\pi^k}} [A_{\pi_\theta}^k(\mathbf{s},a^k)] \tag{12}$$

*Proof.* According to Lemma. (1), we take an arbitrary $k$, cause $\mathbb{E}_{\mathbf{s},a^k\in\mathcal{D}_t^{\pi^k}} \left[A_{\pi_\theta}^k(\mathbf{s},a^k)\right] = 0$, then following (Kuba et al., 2021) we have:

$$\operatorname*{Var}_{\mathbf{s},a^k\in\mathcal{D}_t^{\pi^k}} [A_{\pi_\theta}^k(\mathbf{s},a^k)] = \mathbb{E}_{\mathbf{s},a^k\in\mathcal{D}_t^{\pi^k}} \left[A_{\pi_\theta}^k(\mathbf{s},a^k)^2\right]$$

$$= \mathbb{E}_{\mathbf{s},a^k\in\mathcal{D}_t^{\pi^k}} \left[ \mathbb{E}_{\mathbf{s},\mathbf{a}^{-k}\in\mathcal{D}_t^{\pi^{-k}}} \left[\mathbf{A}_{\pi_\theta}^{1:N}(\mathbf{s},\mathbf{a}^{1:N})\right]^2 \right]$$

$$\leq \mathbb{E}_{\mathbf{s},a^k\in\mathcal{D}_t^{\pi^k}} \left[ \mathbb{E}_{\mathbf{s},\mathbf{a}^{-k}\in\mathcal{D}_t^{\pi^{-k}}} \left[\mathbf{A}_{\pi_\theta}^{1:N}(\mathbf{s},\mathbf{a}^{1:N})^2\right] \right]$$

$$= \mathbb{E}_{\mathbf{s},\mathbf{a}^{-k}\in\mathcal{D}_t^{\pi^{-k}}} \left[ \mathbb{E}_{\mathbf{s},a^k\in\mathcal{D}_t^{\pi^k}} \left[\mathbf{A}_{\pi_\theta}^{1:N}(\mathbf{s},\mathbf{a}^{1:N})^2\right] \right] \tag{13}$$

The above can be equivalently, but more tellingly, rewritten after permuting (cyclic shift) the labels of agents, in the following way

$$\mathbb{E}_{\mathbf{s},\mathbf{a}^{-k}\in\mathcal{D}_t^{\pi^{-k}}} \left[ \mathbb{E}_{\mathbf{s},a^k\in\mathcal{D}_t^{\pi^k}} \left[\mathbf{A}_{\pi_\theta}^{1:N}(\mathbf{s},\mathbf{a}^{1:N})^2\right] \right]$$

$$= \mathbb{E}_{\mathbf{s},\mathbf{a}^{-k}\in\mathcal{D}_t^{\pi^{-k}}} \left[ \mathbb{E}_{\mathbf{s},a^k\in\mathcal{D}_t^{\pi^k}} \left[\mathbf{A}_{\pi_\theta}^{1:k-1,k+1:N,k}(\mathbf{s},\mathbf{a}^{1:k-1,k+1:N,k})^2\right] \right]$$

$$= \mathbb{E}_{\mathbf{s},\mathbf{a}^{-k}\in\mathcal{D}_t^{\pi^{-k}}} \left[ \operatorname*{Var}_{\mathbf{s},a^k\in\mathcal{D}_t^{\pi^k}} \left[\mathbf{A}_{\pi_\theta}^{1:k-1,k+1:N,k}(\mathbf{s},\mathbf{a}^{1:k-1,k+1:N,k})\right] \right], \tag{14}$$

which, by the Lemma. (1), equals

$$\mathbb{E}_{\mathbf{s},\mathbf{a}^{-k}\in\mathcal{D}_t^{\pi^{-k}}} \left[ \operatorname*{Var}_{\mathbf{s},a^k\in\mathcal{D}_t^{\pi^k}} \left[A_{\pi_\theta}^k(\mathbf{s},\mathbf{a}^{-k},a^k)\right] \right]. \tag{15}$$

The equation above can be further simplified by

$$\mathbb{E}_{\mathbf{s},\mathbf{a}^{-k}\in\mathcal{D}_t^{\pi^{-k}}} \left[ \operatorname*{Var}_{\mathbf{s},a^k\in\mathcal{D}_t^{\pi^k}} \left[A_{\pi_\theta}^k(\mathbf{s},\mathbf{a}^{-k},a^k)\right] \right]$$

$$= \mathbb{E}_{\mathbf{s},\mathbf{a}^{-k}\in\mathcal{D}_t^{\pi^{-k}}} \left[ \mathbb{E}_{\mathbf{s},a^k\in\mathcal{D}_t^{\pi^k}} \left[A_{\pi_\theta}^k(\mathbf{s},\mathbf{a}^{-k},a^k)^2\right] \right]$$

$$= \mathbb{E}_{\substack{\mathbf{s},\mathbf{a}^{-k}\in\mathcal{D}_t^{\pi^{-k}} \\ \mathbf{s},a^k\in\mathcal{D}_t^{\pi^k}}} \left[A_{\pi_\theta}^k(\mathbf{s},\mathbf{a}^{-k},a^k)^2\right]$$

$$= \operatorname*{Var}_{\substack{\mathbf{s},\mathbf{a}^{-k}\in\mathcal{D}_t^{\pi^{-k}} \\ \mathbf{s},a^k\in\mathcal{D}_t^{\pi^k}}} \left[A_{\pi_\theta}^k(\mathbf{s},\mathbf{a}^{-k},a^k)\right]. \tag{16}$$

Then we have

$$\operatorname*{Var}_{\substack{\mathbf{s},\mathbf{a}^{-k}\in\mathcal{D}_t^{\pi^{-k}} \\ \mathbf{s},a^k\in\mathcal{D}_t^{\pi^k}}} [A_{\pi_\theta}^k(\mathbf{s},\mathbf{a}^{-k},a^k)] \geq \operatorname*{Var}_{\mathbf{s},a^k\in\mathcal{D}_t^{\pi^k}} [A_{\pi_\theta}^k(\mathbf{s},a^k)], \tag{17}$$

and we sum both sides of the Eq. (17) from $k = 1$ to $N$, we have $\sum_{k=1}^{N} \operatorname*{Var}_{\substack{\mathbf{s},\mathbf{a}^{-k} \in \mathcal{D}_t^{\pi^{-k}} \\ \mathbf{s},a^k \in \mathcal{D}_t^{\pi^k}}} [A_{\pi_\theta}^k(\mathbf{s}, \mathbf{a}^{-k}, a^k)] \geq \sum_{k=1}^{N} \operatorname*{Var}_{\mathbf{s},a^k \in \mathcal{D}_t^\pi} [A_{\pi_\theta}^k(\mathbf{s}, a^k)]$. The proof is finished. $\qquad\square$

## C    METHODOLOGY DETAILS

### C.1    DETAILS OF REGULARIZATION FOR LIPSCHITZ CONTINUITY CONDITION

Lipschitz continuity is essential when employing the Wasserstein loss in various GAN settings (Arjovsky et al., 2017; Xiao et al., 2019). However, recent research (Kim et al., 2021) has demonstrated that the Lipschitz constant of the standard dot product self-attention layer can be unbounded, thereby violating the Lipschitz continuity condition in transformer-based discriminators. We employ two regularization techniques to ensure the Lipschitz continuity condition of our designed discriminator (Lee et al., 2022). Firstly, we adopt L2 attention regularization as proposed in (Kim et al., 2021). It replaces dot product similarity with Euclidean distance and establishes a linkage between the weights of the projection matrices used for querying and self-attention as $\text{Attention}_h(\mathbf{X}) = \text{Softmax}(\frac{d(\mathbf{X}\mathbf{W}_1, \mathbf{X}\mathbf{W}_k)}{\sqrt{d_h}})\mathbf{X}\mathbf{W}_v$, where $\mathbf{W}_q = \mathbf{W}_k$, $\mathbf{W}_q$, $\mathbf{W}_k$, and $\mathbf{W}_v$ are the projection matrices for query, key, and value, respectively. $d(\cdot, \cdot)$ computes vectorized L2 distances between two sets of points. $\sqrt{d_h}$ is the feature dimension for each head. Secondly, we incorporate Spectral Normalization (SN) during discriminator training to further bolster Lipschitz continuity (Miyato et al., 2018). Given that Transformer blocks are sensitive to the Lipschitz constant, and a low Lipschitz constant for MLP blocks can cause the Transformer's output to collapse into a rank-1 matrix (Dong et al., 2021; Lee et al., 2022), we suggest augmenting the spectral norm of the projection matrices to address this concern. Specifically, inspired by (Lee et al., 2022), we multiply the normalized weight matrices of each layer by the spectral norm during initialization and update them according to the following rule for spectral normalization, wherein the standard spectral norm of the weight matrices is computed as $\bar{W}_{SN}(\mathbf{W}) := \sigma(\mathbf{W}_{init}) \cdot \mathbf{W}/\sigma(\mathbf{W})$.

### C.2    DETAILS OF SEQUENTIAL AUTOREGRESSIVE MODELING GENERATOR

Similar to the designed sequential autoregressive discriminator, we also need to consider the problem of modeling complex dependencies among multiple intelligent agents for the generator (policy model). Agents commonly collaborate in various multi-agent cooperative tasks to accomplish a shared objective. Consequently, the efficacy of an agent's actions can be influenced by the actions of teammates, as well as have an impact on their behavior. Neglecting the actions of other agents can result in suboptimal value assessment and hinder effective collaboration. Diverging from existing mechanisms that match individual policies in multi-agent imitation learning, our objective is to construct a generator that incorporates multi-agent interactions to facilitate joint policy matching among intelligent agents. To this end, we propose employing the MAT (Wen et al., 2022) model as the generator, which adeptly models sequences of actions performed by multiple agents in an autoregressive fashion, thereby generating a joint policy distribution for the intelligent agents.

Hence, the generator also adopts "encoder-decoder" architecture, consisting of an encoder that learns representations of the joint observations, and a decoder that outputs actions for each agent in an autoregressive manner. The encoder of the generator exhibits a similar architecture to that of the discriminator, with the distinction that it lacks any regularization mechanisms and includes an additional MLP layer for estimating the individual state values. During the training phase, our objective is to approximate the value function using the encoder and minimize the empirical Bellman error, which can be achieved through the equation:

$$\mathcal{L}_V(\phi) = \frac{1}{T_n} \sum_{k=1}^{N} \sum_{t=0}^{T-1} \left[ r_t^k(\mathbf{s}_t, \mathbf{a}_t^{-k}, a_t^k) + \gamma V_{\bar{\phi}}(\hat{o}_{t+1}^k) - V_\phi(\hat{o}_t^k) \right]^2, \tag{18}$$

where $\bar{\phi}$ is the target network's parameter, which is non-differentiable and updated every few epochs (Wen et al., 2022). In contrast, the decoder of the generator captures the dependencies among the actions of multiple intelligent agents by employing a masked self-attention mechanism. It further incorporates a cross-attention layer to merge the encoder's hidden state representation with

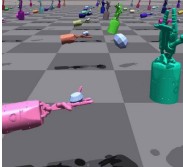 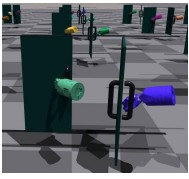 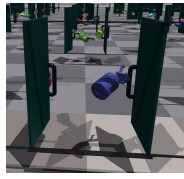 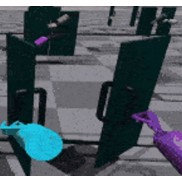 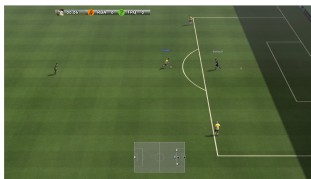

(a) HalfCheetah  (b) CatchOver2Underarm  (c) DoorOpenInward  (d) DoorCloseOutward  (e) DoorCloseInward

Figure 7: Demonstrations of the Bi-DexHands and the HalfCheetah environments.

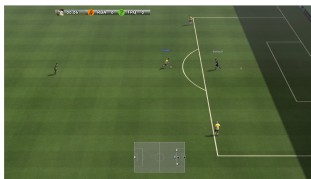 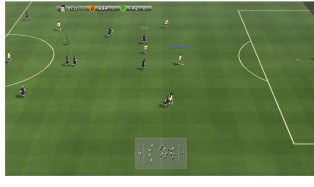 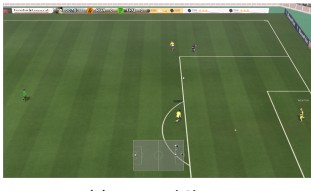

(a) 3 vs 1 with Keeper  (b) Easy Counter-attack  (c) Pass and Shoot

Figure 8: Demonstrations of the Google Football environments.

the intelligent agents' action representation. The first input to the decoder is a symbol denoting the initiation of the decoding process, enabling the generation of the individual action for the first agent. Subsequently, the decoder sequentially produces the complete joint action of the multiple intelligent agents in an autoregressive manner. Following (Wen et al., 2022), to train the decoder, we minimize the following clipping PPO objective of

$$\mathcal{L}_\pi(\theta) = \frac{1}{T_n} \sum_{k=1}^{N} \sum_{t=0}^{T-1} \min(r_t^k(\theta)\hat{A}_t^m, \text{clip}(r_t^k(\theta), 1 \pm \epsilon)\hat{A}_t^k), \tag{19}$$

where $r_t^k(\theta) = \frac{\pi_\theta^k(a_t^k|\mathbf{s}_t, \hat{\mathbf{a}}_t^{1:k-1})}{\pi_{\theta_{old}}^k(a_t^k|\mathbf{s}_t, \hat{\mathbf{a}}_t^{1:k-1})}$, and $\hat{A}_t^k(\mathbf{s}_t, \mathbf{a}_t^{1:k-1}) = r_t^k(\mathbf{s}_t, \mathbf{a}_t^{-k}, a_t^k) + \gamma^N V_\phi(\mathbf{s}_{t+1}, \mathbf{a}_{t+1}) - V_\phi(\mathbf{s}_t, \mathbf{a}_t)$ is an estimate of the individual advantage function.

The proposed framework's training pipeline follows classical adversarial generative imitation learning (Song et al., 2018; Wang et al., 2023), i.e., alternate training for the generator and discriminator according to Eq. (19), Eq. (18), and Eq. (6). We include the MILD$^2$ algorithm as Alg. (1).

# D  EXPERIMENT DETAILS

## D.1  EXPERIMENTAL SETUP

**Benchmark datasets.** We evaluated MILD$^2$ using four benchmarks: StarCraftII Multi-Agent Challenge (SMAC) benchmark (Samvelyan et al., 2019), Google Research Football benchmark (Football) (Kurach et al., 2020), Bimanual Dexterous Hands Manipulation benchmark (Bi-DexHands) (Chen et al., 2022), and Multi-agent MuJoCo benchmark (Ma-Mujoco) (de Witt et al., 2020). We constructed several multi-agent offline datasets on these benchmarks by collecting 10,000 (for tasks in Bi-DexHands) and 100,000 (for tasks in others) transitions of expert policy from HAPPO (Kuba et al., 2022).

- **SMAC**. We constructed an offline dataset using data from the game "StarCraft II" on four maps with discrete action space, each with varying difficulty settings. All maps employ an identical reward function, and the dataset for each map comprises 100,000 transitions.
- **Football**. This benchmark encompasses a series of discrete control tasks in a football game that require cooperation. Our approach was evaluated using data consisting of an average of 100,000 transitions.
- **Bi-DexHands**. This benchmark offers a set of challenging bimanual manipulation tasks: continuous control tasks involving the control of two 24-DoF robotic hands to mimic human behavior. We constructed an offline dataset for three tasks and evaluated our method using 10,000 transitions.

Table 2: Specs of tested tasks (maps) in the SMAC benchmark.

| Name | Agents | Enemies | Type |
|------|--------|---------|------|
| 3m | 3 Marines | 3 Marines | homogeneous & symmetric |
| 3s5z | 3 Stalkers and 5 Zealots | 3 Stalkers and 5 Zealots | heterogeneous & asymmetric |
| 6h vs 8z | 6 Hydralisks | 8 Zealots | micro-trick: focus fire |
| MMM2 | 1 Medivac, 2 Marauders & 7 Marines | 1 Medivac, 3 Marauders & 8 Marines | heterogeneous & asymmetric |

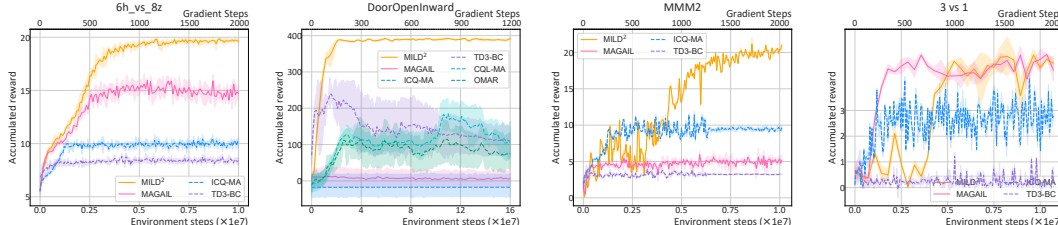

Figure 9: The learning curve comparisons on several complex and challenging tasks from SMAC, Football, and Bi-DexHands benchmarks. MILD[2] and MAGAIL use environmental steps as the horizontal axis, and other non-GAIL methods use gradient steps.

- **Ma-Mujoco**. This benchmark comprises a collection of continuous control tasks in machine learning. In each Ma-MuJoCo environment, each agent controls a specific part of a shared robot (e.g., a leg of a Hopper), and all agents aim to maximize a shared reward function. Our method was evaluated using 100,000 transitions.

We adopt the open-source implementations for these Benchmarks. Fig. (7) and Fig. (8) illustrate tasks from these benchmarks. The *HalfCheetah 6×1* task is shown in Fig. (7a) while Fig. (7b-e) illustrate the *CatchOver2Underarm*, *DoorOpenInward*, *DoorCloseOutward* and *DoorCloseInward* environments from the Bi-DexHands benchmark. Fig. (8a-c) illustrate the *3 vs 1 with Keeper*, *Easy Counter-attack*, and *Pass and Shoot* environments from the Football benchmark. Tested maps in the StarCraft II micromanagement benchmark are summarized in Tab. (2).

**Baselines.** We compare our method against five classical multi-agent offline RL methods, including MAGAIL (Song et al., 2018), the multi-agent version of CQL (CQL-MA) (Kumar et al., 2020), ICQ-MA (Yang et al., 2021), TD3-BC (Fujimoto & Gu, 2021), and OMAR (Pan et al., 2022). Following most baseline methods, each algorithm runs with five seeds, where the performance is evaluated 20 times every 50 episodes.

**Implementation Details.** All baseline methods were implemented consistently with their official repositories, maintaining their hyper-parameters at their original best-performing settings. In our approach, we employed a 1-block Transformer-based discriminator and a 1-block Transformer-based generator for all tasks. Following MAT (Wen et al., 2022), the feature dimension for all Transformer blocks and MLP layers was set to 64. The learning rates for the actors and critics were set to 5e-4, 5e-4, 5e-5, and 5e-5 for the SMAC, Football, Bi-DexHands, and Ma-Mujoco benchmarks, respectively. The training of our models was conducted on a single NVIDIA Tesla V100 GPU. The batch size and update epoch for updating the generator and discriminator once were set to 128 and 5.

## D.2 ABLATION STUDIES

**Analysis on different distribution matching settings.** We comprehensively analyzed three variants of the MILD[2] model to evaluate the effects of incorporating global dependencies on the discriminator and generator components. As shown in Tab. (3), a substantial deterioration in the model's performance was observed when the global dependencies were eliminated from the discriminator (i.e., independent modeling of individual reward distributions). Likewise, a significant decline in performance was observed when the global dependencies were removed from the generator (i.e., independently modeling individual policy distributions). These empirical findings emphasize the indispensability of introducing global dependencies and establishing joint reward and policy distributions within the model.

Table 3: Performance comparison across different distribution matching settings to evaluate the effect of introducing global dependencies on the discriminator and generator, where MILD$^2$ is the original implementation; Ind. Disc involves an independent discriminator (reward function) without global dependencies; Ind. Gen utilizes independent generator (individual policies) without global dependencies; Ind. D & G combines both independent discriminator and independent generator. The winning rates are shown in brackets.

| Tasks | MILD$^2$ | MILD$^2$ (Ind. Disc) | MILD$^2$ (Ind. Gen) | MILD$^2$ (Ind. D & G) |
|---|---|---|---|---|
| 3m | **20.00±0.00** **(1.00±0.00)** | 19.91±0.06 (0.99±0.01) | 18.88±0.06 (0.99±0.01) | 19.97±0.05 (0.99±0.01) |
| 3s5z | **20.00±0.02** **(1.00±0.00)** | 19.80±0.08 (0.95±0.02) | 19.89±0.08 (0.97±0.02) | 19.92±0.02 (0.94±0.02) |
| 6h vs 8z | **19.78±0.07** **(0.96±0.01)** | 19.54±0.14 (0.88±0.02) | 17.10±0.24 (0.49±0.05) | 16.82±0.29 (0.47±0.02) |
| MMM2 | **20.52±0.09** **(0.86±0.03)** | 5.39±0.07 (0.00±0.00) | 20.21±0.15 (0.85±0.12) | 5.01±0.03 (0.00±0.00) |

Table 4: Performance comparison for different discriminator and generator architectures to explore the effect of each component, where MILD$^2$ is the original implementation; MILD$^2$-dec is implemented with the encoder only, without the autoregressive process; MILD$^2$-enc is implemented with the decoder only, keeping the auto-regressive process; MILD$^2$+GRU is implemented with GRU instead of Transformer for modeling global dependencies.

| Tasks | MILD$^2$ | MILD$^2$-Dec | MILD$^2$-Enc | MILD$^2$+GRU |
|---|---|---|---|---|
| 3s5z | **20.00±0.02** **(1.00±0.00)** | 19.81±0.08 (0.95±0.02) | 19.93±0.03 (0.98±0.00) | 19.95±0.02 (0.98±0.00) |
| 6h vs 8z | **19.78±0.07** **(0.96±0.01)** | 16.22±0.21 (0.39±0.03) | 19.57±0.16 (0.92±0.02) | 19.63±0.11 (0.93±0.03) |
| MMM2 | **20.52±0.09** **(0.86±0.03)** | 4.87±0.03 (0.00±0.00) | 18.29±0.35 (0.64±0.03) | 18.93±0.18 (0.72±0.05) |
| DoorOpenInward | **395.98±0.39** | 382.31±0.05 | 386.23±0.78 | 391.08±0.46 |

**Analysis on different model architectures.** We evaluated several model variants to assess the contribution of each architectural component. Our ablation study, conducted in both homogeneous and heterogeneous settings, also tested a GRU-based discriminator in place of the Transformer (Tab. (4). All the simplified variants exhibited slower convergence and marginally lower performance, demonstrating that modeling global dependencies with a Transformer is critical. The full Transformer encoder–decoder consistently outperformed all alternatives, underscoring its importance to our method.

## D.3 ROBUSTNESS STUDIES

### D.3.1 NOISY DATA REGIME

In this section, we endeavor to substantiate our hypothesis, positing that the sequential autoregressive policy and reward models exhibit enhanced robustness. This heightened robustness stems from the model's action evaluation and decision-making processes contingent upon global correlations rather than solely relying on localized information. Even in scenarios where datasets encompass a degree of noisy transitions, the model continues to demonstrate commendable performance. This capability to harness global correlations proves particularly advantageous in settings characterized by non-Markovian dynamics, such as cooperative tasks, wherein the decisions made by other agents wield influence over future outcomes. To empirically investigate the validity of our hypothesis, we created a "mixed" dataset through the amalgamation of medium datasets (exploring with medium-score policies), deliberately introduced as sources of noise, with an expert dataset. This amalgamation comprises datasets featuring varying noise ratios. Subsequently, we subjected the proposed sequential autoregressive framework MILD$^2$, along with the baseline models MAGAIL and ICQ-MA, to a battery of tests under diverse noise ratios, as illustrated in Fig. (10). The results consistently demonstrated that MILD$^2$ outperformed MAGAIL and ICQ-MA across all experimental configurations. Notably,

Table 5: Mean accumulated trajectory rewards with standard deviation across baselines on four benchmarks.

| Benchmarks | Tasks | MAGAIL | CQL-MA | ICQ-MA | TD3-BC | OMAR | MILD²(Ours.) |
|---|---|---|---|---|---|---|---|
| SMAC | 3m | 19.97±1.56 | 15.20±1.93 | 12.87±1.56 | 12.64±1.40 | 19.82±0.02 | **20.00±0.00** |
| | 3s5z | 19.92±0.02 | 10.66±0.93 | 18.77±0.50 | 14.40±1.09 | 19.87±0.11 | **20.00±0.02** |
| | 6h vs 8z | 16.90±0.12 | 7.91±0.14 | 10.13±0.35 | 8.56±0.18 | 16.33±0.15 | **19.78±0.07** |
| | MMM2 | 5.01±0.03 | 5.01±0.03 | 11.32±0.87 | 3.09±0.24 | 10.99±0.22 | **20.52±0.09** |
| Football | 3 vs 1 | 4.88±0.04 | 4.18±0.47 | 2.85±0.58 | 1.42±0.33 | 4.55±0.34 | **4.89±0.02** |
| | counterattack | 4.62±0.08 | 0.36±0.02 | 2.21±0.44 | 0.27±0.06 | 1.14±0.18 | **4.77±0.16** |
| | pass and shoot | 3.92±0.39 | 1.39±0.71 | 3.11±0.62 | 1.68±0.09 | 2.72±0.58 | **4.83±0.11** |
| Bi-DexHands | CatchOver2Underarm | 6.70±0.10 | 15.65±1.01 | 3.53±0.17 | 10.13±1.15 | 16.85±1.21 | **24.16±0.62** |
| | DoorOpenInward | 12.47±23.34 | 189.74±41.71 | -7.20±36.08 | 217.68±45.01 | 114.47±34.31 | **395.98±0.39** |
| | DoorCloseOutward | 503.32±0.12 | 839.48±11.29 | 215.365±0.08 | 41.84±5.16 | 818.76±2.43 | **1016.89±0.13** |
| Multi-agent Mujoco | HalfCheetah 6×1 | 435.46±9.00 | 2189.50±959.35 | 3977.20±127.14 | 4123.60±146.41 | 4088.93±165.67 | **4475.95±74.75** |

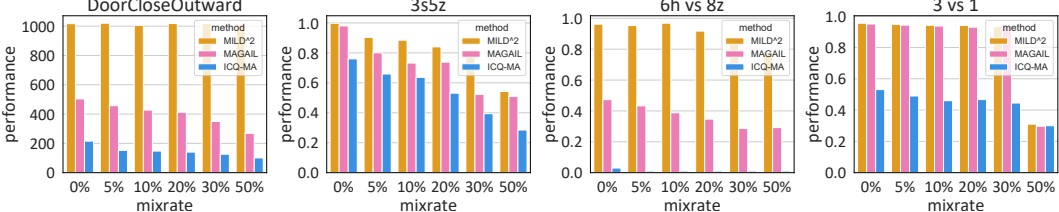

Figure 10: Performance comparison of various methods in noisy data regimes. In the leftmost environment (from the Bi-DexHands benchmark), cumulative rewards are employed as the performance evaluation metric, whereas in the remaining environments (from SMAC and Football benchmark), win rates serve as the performance evaluation metric.

the performance of both MAGAIL and ICQ-MA exhibited conspicuous susceptibility to variations in the noise ratio, showing a precipitous decline as the noise ratio increased from 0% to 50%. In contrast, MILD² performed well even when the noise ratio reached 50%. For instance, in the 6h vs 8z scenario, MILD² maintained a win rate of approximately 90%, even in the presence of a 50% noise ratio. Conversely, in the 3s vs 5z scenario, MAGAIL and ICQ-MA displayed vulnerability to noise, even at lower noise ratios such as 5%.

### D.3.2 SMALL DATA REGIME

This section aims to investigate the advantages of capturing global dependencies for in-sample generalization using MILD². Precisely, we assess its robustness compared to baseline models when dealing with limited dataset size or reduced dataset diversity in specific states, common challenges in imitation learning with real-world data. To conduct this investigation, we curated custom datasets by excluding specific transitions in datasets like DoorCloseOutward, 3s5z, 6h vs 8z, and counterattack. The exclusion criteria are based on the proximity to the target location, simulating scenarios where data near the task goal is constrained due to the stochastic nature of data generation policies (Xu et al., 2023). We introduced a retention ratio parameter governing dataset composition to simulate demonstration data at different scales. We compared the performance of MILD² to two other models, MAGAIL and ICQ-MA, by measuring average accumulated trajectory rewards (win rate) during evaluation and training standard deviation, as shown in Tab. (6). MAGAIL and ICQ-MA experience a significant drop in performance as the retention ratio decreases. In more challenging tasks, the standard deviation increases exponentially, indicating a substantial growth in generalization error with limited data. In contrast, MILD² consistently demonstrates stable and commendable performance across various retention ratios. Additionally, MILD² has a notably more minor standard deviation than MAGAIL and ICQ-MA. This compelling evidence highlights the advantage of capturing global dependencies, enabling better use of dataset samples to mitigate incorrect value estimations and improve overall performance.

### D.4 ANALYSIS OF COOPERATION SCALE

As shown in Tab. (1), MILD² performs better in tasks with large cooperation scales due to its ability to model global interdependencies among agents and increased variance of advantage actions.

Table 6: The average accumulated trajectory rewards (win rates) and standard deviation of MILD$^2$, MAGAIL, and ICQ-MA in small data regimes.

| Benchamarks | | Bi-Dexhands | SMAC | | | | Football | |
|---|---|---|---|---|---|---|---|---|
| Tasks | | DoorCloseOutward | 3s5z | | 6h vs 8z | | counterattack | |
| Method | ratio | acc. rewards | acc. rewards | win. rate | acc. rewards | win. rate | acc. rewards | win. rate |
| MILD$^2$ | 1% | 1015.92±0.13 | 9.09±0.19 | (0.00±0.00) | 4.13±0.08 | (0.00±0.00) | 4.70±0.19 | (0.90±0.04) |
| | 5% | 1015.84±0.11 | 18.87±0.21 | (0.79±0.03) | 19.00±0.11 | (0.74±0.03) | 4.75±0.19 | (0.91±0.08) |
| | 10% | 1014.93±0.12 | 19.76±0.10 | (0.94±0.01) | 19.55±0.16 | (0.93±0.03) | 4.83±0.10 | (0.93±0.02) |
| | 20% | 1015.74±0.12 | 19.85±0.09 | (0.96±0.02) | 19.77±0.16 | (0.96±0.03) | 4.89±0.12 | (0.94±0.03) |
| | 40% | 1016.06±0.13 | 19.94±0.04 | (0.98±0.01) | 19.74±0.13 | (0.96±0.02) | 4.85±0.09 | (0.93±0.02) |
| MAGAIL | 1% | 370.72±0.37 | 14.33±0.34 | (0.29±0.04) | 9.45±0.21 | (0.00±0.00) | 4.44±0.06 | (0.85±0.01) |
| | 5% | 445.81±0.28 | 19.47±0.17 | (0.89±0.03) | 12.24±0.37 | (0.11±0.03) | 4.50±0.13 | (0.85±0.03) |
| | 10% | 463.75±0.20 | 19.68±0.13 | (0.89±0.03) | 13.42±0.27 | (0.17±0.04) | 4.58±0.18 | (0.87±0.04) |
| | 20% | 480.00±0.23 | 19.82±0.11 | (0.96±0.03) | 14.82±0.42 | (0.32±0.04) | 4.58±0.25 | (0.88±0.06) |
| | 40% | 491.36±0.16 | 19.84±0.08 | (0.96±0.02) | 15.69±0.55 | (0.35±0.08) | 4.64±0.06 | (0.89±0.01) |
| ICQ-MA | 1% | 149.74±0.21 | 13.25±0.45 | (0.02±0.00) | 8.09±0.22 | (0.00±0.00) | 0.98±0.19 | (0.14±0.08) |
| | 5% | 189.97±0.12 | 15.03±0.62 | (0.17±0.09) | 9.42±0.21 | (0.00±0.00) | 1.57±0.49 | (0.26±0.09) |
| | 10% | 202.61±0.09 | 16.24±0.69 | (0.35±0.11) | 9.37±0.30 | (0.00±0.00) | 1.78±0.59 | (0.29±0.12) |
| | 20% | 216.46±0.14 | 17.50±0.52 | (0.49±0.09) | 9.38±0.38 | (0.00±0.00) | 2.02±0.55 | (0.36±0.18) |
| | 40% | 177.30±0.12 | 18.29±0.60 | (0.63±0.12) | 9.88±0.44 | (0.01±0.02) | 1.15±0.28 | (0.20±0.05) |

For example, in the case of SMAC, agents' scalability is demonstrated across four tasks: 3m, 3s5z, 6h vs 8z, and MMM2, with scalability factors of 3, 8, 8, and 12, respectively. MILD$^2$ significantly outperforms baseline methods like MAGAIL, with an 81.27% improvement, especially in the expansive cooperative setting of MMM2. However, in smaller-scale environments like 3m, the improvement is only 0.15%.

These findings highlight that larger cooperative environments exhibit complex agent dependency structures. MILD$^2$'s ability to capture and represent these global dependencies is crucial for accurately modeling cooperative relationships and allocating credit among agents. It helps reduce distribution matching errors caused by the non-stationarity of environmental dynamics.

# E    HYPERPARAMETERS

During experiments, the implementations of baseline methods are consistent with their official repositories, and all hyper-parameters left unchanged at the origin best-performing status. The hyperparameters adopted for the discriminator are listed in Tab. (7), and those adopted for the generator are listed in Tab. (8-12).

Table 7: Hyperparameters used for the discriminator (reward model) in four benchmarks.

| Benchmarks | Hyper-Parameter | Default Configuration |
|---|---|---|
| Common parameters | optimizer | Adam |
| | scheduler | StepLR |
| | hidden_size | 64 |
| | batch size | 128 |
| | learning rate | 5e-6 |
| | disc epoch | 5 |
| | disc warmup epoch | 100 |
| | disc warmup steps | 10 |
| | layers num | 1 |
| | heads num | 1 |
| Bi-DexHands | max grad norm | 0.5 |
| | distance metric | Wasserstein distance |
| | attention type | Dot-product attention |
| SMAC | max grad norm | 10.0 |
| | distance metric | KL divergence |
| | attention type | L2 attention |
| Football | max grad norm | 0.5 |
| | distance metric | KL divergence |
| | attention type | L2 attention |
| Ma-MuJoCo | max grad norm | 0.5 |
| | distance metric | Wasserstein distance |
| | attention type | Dot-product attention |

Table 8: Common hyperparameters used for the generator (policy model) in the SMAC benchmark.

| Hyperparameter | Value | Hyperparameter | Value | Hyperparameter | Value |
|---|---|---|---|---|---|
| critic lr | 5e-4 | actor lr | 5e-4 | use gae | TRUE |
| gain | 0.01 | optim eps | 1e-5 | batch size | 3200 |
| training threads | 16 | num mini-batch | 1 | rollout threads | 32 |
| entropy coef | 0.01 | max grad norm | 10 | episode length | 100 |
| optimizer | Adam | hidden layer dim | 64 | use huber loss | TRUE |

Table 9: Different hyperparameters used for the generator (policy model) in the SMAC benchmark.

| Tasks | ppo epochs | ppo clip | num blocks | num heads | stacked frames | steps | $\gamma$ |
|---|---|---|---|---|---|---|---|
| 3m | 15 | 0.2 | 1 | 1 | 1 | 5e5 | 0.99 |
| 3s5z | 10 | 0.05 | 1 | 1 | 1 | 3e6 | 0.99 |
| 6h vs 8z | 15 | 0.05 | 1 | 1 | 1 | 1e7 | 0.99 |
| MMM2 | 5 | 0.05 | 1 | 1 | 1 | 1e7 | 0.99 |

Table 10: Hyperparameters used for the generator (policy model) in the Bi-Dexhands benchmark.

| Hyperparameter | Value | Hyperparameter | Value | Hyperparameter | Value |
|---|---|---|---|---|---|
| cirtic lr | 5e-5 | actor lr | 5e-5 | hidden dim | 64 |
| gamma | 0.96 | steps | 5e7 | stacked frames | 1 |
| gain | 0.01 | optim eps | 1e-5 | ppo epochs | 5 |
| ppo clip | 0.2 | num mini-batch | 1 | rollout threads | 80 |
| batch size | 6000 | episode length | 75 | optimizer | Adam |
| entropy coef | 0.001 | max grad norm | 0.5 | training threads | 16 |

Table 11: Hyperparameters used for the generator (policy model) in the Football benchmark.

| Hyperparameter | Value | Hyperparameter | Value | Hyperparameter | Value |
|---|---|---|---|---|---|
| cirtic lr | 5e-4 | actor lr | 5e-4 | gamma | 0.99 |
| ppo clip | 5e-2 | num head/block | 1 | ppo epochs | 10 |
| gain | 0.01 | optim eps | 1e-5 | batch size | 4000 |
| training threads | 16 | num mini-batch | 1 | rollout threads | 20 |
| entropy coef | 0.001 | max grad norm | 0.5 | episode length | 200 |
| optimizer | Adam | hidden layer dim | 64 | stacked frames | 1 |

Table 12: Hyperparameters used for the generator (policy model) in the Ma-Mujoco benchmark.

| Hyperparameter | Value | Hyperparameter | Value | Hyperparameter | Value |
|---|---|---|---|---|---|
| cirtic lr | 5e-5 | actor lr | 5e-5 | ppo epochs | 10 |
| ppo clip | 5e-2 | num block | 1 | num head | 1 |
| gamma | 0.99 | steps | 1e7 | stacked frames | 1 |
| gain | 0.01 | optim eps | 1e-5 | batch size | 4000 |
| training threads | 16 | num mini-batch | 40 | rollout threads | 40 |
| entropy coef | 0.001 | max grad norm | 0.5 | episode length | 100 |
| optimizer | Adam | hidden layer dim | 64 | use huber loss | TRUE |