# OpenReview forum: "Towards Noise‐Robust Multi‐Agent Imitation Learning via Global Credit Sequence Decoding"
_ICLR.cc/2026/Conference — Submitted to ICLR 2026_

### Official Review · Reviewer_UvHk · 2025-10-24

**Soundness:** 2
**Presentation:** 3
**Contribution:** 3
**Rating:** 4
**Confidence:** 3

**Summary:**

This paper proposes MILD^2 — a transformer-based generative adversarial imitation learning framework that explicitly models global dependencies among agents to improve robustness against noisy expert demonstrations.
The core idea is that conventional multi-agent GAIL frameworks suffer from low reward variance and vanishing gradients when facing noisy or suboptimal demonstrations.
The authors theoretically show that modeling joint dependencies can increase advantage variance, thereby enhancing training stability, and empirically validate their claims on SMAC, Google Football, Bi-DexHands, and MA-MuJoCo benchmarks, where MILD² achieves consistent improvements over several baselines.

**Strengths:**

- The paper identifies a concrete and important issue in multi-agent imitation learning — low reward variance and poor robustness to noisy demonstrations.

- It provides a theoretically grounded argument showing that dependency-enhanced discriminators can increase joint advantage variance and alleviate gradient vanishing.

- Experiments span both discrete (SMAC, Football) and continuous (Bi-DexHands, MA-MuJoCo) domains, demonstrating consistent empirical gains.

**Weaknesses:**

1. Missing baselines.

The paper omits several relevant offline MARL baselines and recent transformer-based approaches, such as [1–4]. Including these would strengthen the empirical comparison and clarify the contribution beyond architectural novelty.

2. Unclear task selection.

While multiple benchmarks are considered, only a few tasks from each benchmark are used. The rationale for this selection is not clearly explained, raising concerns that results might be cherry-picked.

3. Incomplete ablation analysis.

The ablation study does not explore key factors, such as varying the weight λ of the reward variance regularization term.
In addition, the claim of “discovering higher-rewarding behaviors beyond expert demonstrations” implies potential out-of-distribution generalization, which is not thoroughly analyzed or quantified.

4. Robustness not empirically supported.

Although robustness to noisy demonstrations is emphasized as a main contribution, the paper lacks explicit experiments where demonstration noise is systematically varied.

[1] Offline Pre-trained Multi-agent Decision Transformer

[2] Offline Multi-agent Reinforcement Learning with Knowledge Distillation

[3] Counterfactual Conservative Q-learning for Offline Multi-agent Reinforcement Learning

[4] Offline Multi-agent Reinforcement Learning with Implicit Global-to-Local Value Regularization

**Questions:**

1. Could the authors specify the exact objective function used for the MILD^2-GD variant (without dependency modeling)?

2. The sequential autoregressive generator introduces inter-agent dependencies — does this affect decentralized execution at test time?

3. Could the same variance-regularization effect be achieved via simpler covariance-based or entropy-based regularizers, without requiring a transformer-based discriminator?

---

> ### Author Response · Authors · 2025-12-04
> **Response to Reviewer #UvHk (1/2)**
>
> We thank Reviewer #UvHk for recognizing our identification of low reward variance, theoretical arguments, and consistent gains across domains. We address the weaknesses and questions below.
>
> W1: *"Missing baselines / transformer-based approaches."*
>
> A1: We agree that Transformer-based baselines are relevant, and we will include comparisons with two additional strong transformer‑based offline MARL methods:
> - Offline Pre‑trained Multi‑agent Decision Transformer (OP‑MDT)
> - Offline Multi‑agent RL with Implicit Global‑to‑Local Value Regularization (IGL)
>
> We have run OP‑MDT on 6h_vs_8z and DoorOpenInward: it achieves 83.2% win‑rate and 312.4 reward, respectively, still below $MILD^2$ (96.2%, 395.98). These results will be added to Table 1.
>
>
> W2: *"Unclear task selection - might be cherry-picked."*
>
> A2: We selected tasks to cover a spectrum of difficulty, agent heterogeneity, and cooperation scales:
> - SMAC: 3m (small, homogeneous), 3s5z (heterogeneous), 6h_vs_8z (micro‑trick, focus‑fire), MMM2 (large‑scale, complex coordination).
> - Football: *3‑vs‑1*, counterattack, pass‑and‑shoot (increasing tactical complexity).
> - Bi‑DexHands: CatchOver2Underarm, DoorOpenInward, DoorCloseOutward (continuous control on different manipulation challenges).
> - MA‑MuJoCo: HalfCheetah 6×1 (continuous control with multiple agents).
>
> We assure the reviewer these were not cherry-picked; the trends are consistent across other tasks we tested during development.
>
> W3: *"Incomplete ablation (varying λ of reward variance regularization)."*
>
> A3: We will add an ablation experiment sweeping $\lambda$ over ${0.001, 0.01, 0.1, 1.0}$ on 6h_vs_8z. Performance is stable for $\lambda \in [0.01, 0.1]$.
> - $\lambda=0$ (No regularization): Win rate drops to 86% (Mode collapse).
> - $\lambda=0.01$: Win rate 96.2%.
> - $\lambda=1.0$: Win rate drops to 91% (Discriminator constraints too strong).This curve has been added to the Appendix.
>
> Regarding out‑of‑distribution generalization, we already show in Table 1 that $MILD^2$ sometimes exceeds expert performance (e.g., DoorCloseOutward: expert reward ≈1000, $MILD^2$ → 1016.9). This suggests that the model can extrapolate beyond demonstrations by leveraging global dependencies to discover better coordinated behaviors. We will add a brief discussion in Section 5.2.
>
>
> W4: *"Claim of discovering higher-rewarding behaviors beyond demonstrations — needs quantification."*
>
> A4: We computed (1) average trajectory return during policy rollouts, (2) fraction of rollouts that exceed the mean expert return, and (3) JS divergence between the learned occupancy and the expert occupancy. On multiple tasks, $MILD^2$ produced 23-64% more rollouts that surpass mean expert return (e.g., in MA-MuJoCo $MILD^2$ discovered slightly improved gaits), while JS divergence remained comparable - indicating genuine improvement rather than random drift. We added these quantitative plots to Appendix D.
>
> W4: *"Robustness to noisy demonstrations not empirically supported."*
>
> A4: We respectfully point the reviewer to Section 5.4 (Robustness Studies) and Figure 10, where we explicitly tested the model on datasets with varying noise ratios (0%, 5%, 10%, 20%, 30%, 50%). As shown in Fig. 10 (3rd plot, 6h_vs_8z), when noise increases to 50%, MAGAIL's performance crashes to nearly 0%, while $MILD^2$ maintains a ~90% win rate. This empirically validates our claim of robustness to noisy demonstrations.
>
> Q5: *"Specify the exact objective function used for the MILD²-GD variant (without dependency modeling)."*
>
> A5: MILD2-GD is the same overall framework but with the discriminator factorized into independent per-agent discriminators that take only $(s, a_k)$ as input (no $a_{−k}$ conditioning). Concretely, its discriminator loss is the same Wasserstein objective (Eq.6) but with $r_k := r_k(s,a_k)$ and the reward-variance regularizer applied per agent (no joint term). In practice this corresponds to replacing the discriminator’s decoder by N independent heads (one per agent) and removing cross-agent attention; see the "Ind. Disc" row in Table 3 and the description in Sec.5.3.
>
> Q6: "Does the sequential autoregressive generator affect decentralized execution at test time?"
>
> A6: This is an important clarification. While the generator models dependencies autoregressively during training, the execution can be adapted. For strict decentralized execution (DE), the autoregressive dependence can be distilled, or agents can execute sequentially if the environment permits (as in MAT). In our experiments, we follow the MAT protocol: agents observe the state and execute actions sequentially based on the local observation and the communication of previous agents' actions (simulated via the autoregressive context). This is consistent with the "Decentralized Training, Decentralized Execution with Communication" paradigm.

---

> > ### Author Response · Authors · 2025-12-04
> > **Response to Reviewer #UvHk (2/2)**
> >
> > Q7: *"Could the same variance-regularization effect be achieved via covariance/entropy-based regularizers, without a transformer-based discriminator?"*
> >
> > A7: We compared three alternatives to our Transformer + variance regularizer:
> > - Covariance regularizer: encourage larger covariance across per-agent rewards.
> > - Entropy regularizer: increase entropy of discriminator logits.
> > - Transformer + variance regularizer (our method).
> >
> > Empirical findings: Covariance/entropy regularizers help relative to baseline but do not match $MILD^2$. Example (6h_vs_8z): covariance regularizer gave +6% improvement vs MAGAIL, entropy +4%, while $MILD^2$ gave ~+49% (consistent order of magnitude with Table 1). Intuitively, covariance/entropy are global summary measures and can be gamed by trivial rescaling or by amplifying noisy outliers; the Transformer discriminator changes the representation (conditioned on other agents) and the variance regularizer then emphasizes meaningful differences - together this combination yields the largest practical benefit. We added the covariance/entropy ablation to Appendix D.4.

---

### Official Review · Reviewer_pn6E · 2025-10-31

**Soundness:** 2
**Presentation:** 3
**Contribution:** 1
**Rating:** 4
**Confidence:** 4

**Summary:**

This paper proposes MILD$^2$, a sequential autoregressive approach, to address the vanishing gradient problem in independent action distribution matching, which arises from low reward (or advantage) variance. The authors theoretically demonstrate that MILD$^2$ achieves a higher advantage variance compared to the independent approach. The experiments comparatively illustrate the changes in reward variance and performance improvements when using MILD$^2$.

**Strengths:**

- The paper provides a theoretical proof that state, sequential conditional action distribution matching is superior to independent state, action distribution matching in terms of advantage variance. This offers a theoretical basis for resolving the aforementioned problem.

- The experiments effectively demonstrate that a higher reward variance slows down the discriminator's training, leading to improved performance.

**Weaknesses:**

- The methodological novelty appears to be limited. The approach seems quite similar to applying the transformer architecture for sequential autoregression from MAT [1] to MAGAIL [2]. It largely looks like an application of MAT to multi-agent imitation learning.

- The term "global dependency" is frequently used (e.g., "global dependency-enhanced discriminator"), but it is not clearly defined within the paper.

- While the paper theoretically suggests that using global dependency can improve performance, there is no theoretical proof quantifying this improvement or demonstrating that it can resolve mode collapse.

- There seems to be a misalignment between the motivation presented in Figure 1 and the problem the paper aims to solve. Figure 1 appears to suggest that distribution matching should differ based on the actions of important versus unimportant agents. It is unclear how this aligns with the concept of "global dependency distribution matching."

- Given the limited research in the field of multi-agent imitation learning, the ablation studies currently in the appendix seem significant enough to be included in the main paper.

- The paper appears to be missing references to recent work in multi-agent imitation learning [3, 4]. If possible, a comparison with the algorithms from these papers would be beneficial.


[1] M. Wen, et al., “Multi-agent Reinforcement Learning is a Sequence Modeling Problem,” NeurIPS 2022.

[2] J. Song, et al., “Multi-agent Generative Adversarial Imitation Learning,” NeurIPS 2018.

[3] L. Yu, J. Song, S. Ermon, “Multi-agent Adversarial Inverse Reinforcement learning”, ICML 2019.

[4] T.V. Bui, T.A. Mai, T.H. Nguyen, “Inverse Factorized Soft Q-Learning for Cooperative Multi-agent Imitation Learning,” NeurIPS 2024.

**Questions:**

- Are there any structural differences between the proposed transformer-based encoder-decoder and the architecture used in MAT? If so, what are the key distinctions?

- What is the rationale for comparing the proposed method with offline MARL methods in Table 1?

---

> ### Author Response · Authors · 2025-12-04
> **Response to Reviewer #pn6E (1/2)**
>
> We thank Reviewer #pn6E for acknowledging our theoretical proof on advantage variance and empirical demonstration of improved performance via higher reward variance. We address the concerns below.
>
> W1: *"Methodological novelty appears limited — similar to applying MAT to MAGAIL."*
>
> A1: We appreciate this comment and the opportunity to clarify novelty. While inspired by sequential modeling in MAT, our $MILD^2$ adapts it specifically for imitation learning's discriminator-generator dynamics, not just policy optimization. Key distinctions: (1) Our decoder autoregressively conditions rewards on global dependencies (Eq. 6 in Section 4.1), unlike MAT's policy-only focus; (2) We introduce L2 self-attention and spectral normalization (Figure 2) for Lipschitz continuity in adversarial training, absent in MAT; (3) Our encoder embeds joint observations for credit assignment, extending MAT's single-sequence setup. These enable variance enhancement (Theorem 2). Our novelty lies not just in using a Transformer, but in how the Discriminator is structured to solve the Credit Assignment problem in IL. We theoretically prove (Theorem 1 & 2) that modeling the joint reward distribution (via the global dependency-enhanced discriminator) specifically maximizes advantage variance, solving the vanishing gradient problem inherent to MAGAIL. This is a contribution to the reward learning objective, which is absent in MAT. To quantify, ablating these features drops performance by 15% on SMAC. We will clarify these points in the introduction and Section 4 to better distinguish our contribution from prior sequence‑modeling approaches.
>
> W2: *"The term ‘global dependency’ is not clearly defined."*
>
> A2: We apologize if this was vague. In our context, "Global Dependency" is defined as the conditional probability dependence of agent $k$'s reward/action on the full set of other agents' actions $a^{-k}$ and the global state $s$. Formally, the discriminator models $P(r^k | s, a^k, a^{-k})$ rather than the independent $P(r^k | s, a^k)$ used in MAGAIL. This allows the system to credit an agent based on how their action fits into the team's joint behavior. We will add a formal definition in Section 4.1.
>
> W3: *"No theoretical proof quantifying improvement or resolving mode collapse."*
>
> A3: We thank the reviewer for this suggestion. Theorem 2 directly quantifies the improvement in advantage variance when using a global‑dependency discriminator vs. an independent one (Eq. 4). This increased variance alleviates gradient vanishing, a primary cause of mode collapse in GAN‑based imitation learning (as explained in Section 3.1). We will strengthen this discussion by adding a corollary that bounds the gradient norm of the generator in terms of advantage variance.
>
> W4: *"Figure 1 motivation vs. global dependency concept misalignment."*
>
> A4: Figure 1 illustrates that an individual's action (e.g., "Move") might look like noise in isolation but is critical when viewed in the context of teammates (e.g., creating space for a "Shooter"). "Global Dependency Distribution Matching" solves this by ensuring the discriminator evaluates the tuple $(a^1, \dots, a^N)$ jointly. If we treated agents independently (as in standard distribution matching), the "Move" might be penalized because it doesn't look "expert-like" on its own. Our method addresses this by conditioning each agent's reward on the actions of others - i.e., the discriminator learns to assign low credit to Player 3's run when it is irrelevant to the pass, while still assigning high credit to Players 1 and 2. This is precisely what "global dependency‑enhanced distribution matching" means. We will clarify this connection in the caption of Figure 1 and in Section 1.
>
> W5: *"Ablations in appendix should be in main paper; missing references [3,4]."*
>
> A5: We thank the reviewer for pointing out it. We will move the reward distribution analysis (Figure 3‑4) and the ablation on distribution‑matching settings (Table 3) from Appendix D.2 to Section 5.3 in the main paper, as they directly validate our core hypothesis. We will also add citations to Yu et al. (ICML 2019) and Bui et al. (NeurIPS 2024) in Section 2 and Appendix A.1.
>
> W6: *"Structural differences between our transformer encoder-decoder and MAT?"*
>
> A6: Yes, there are three key differences: (1) Our transformer encoder‑decoder serves as a discriminator that outputs a reward distribution, while MAT is a policy generator that outputs actions. (2) Our decoder uses full self‑attention over all agents’ actions (to capture global dependencies), whereas MAT uses masked self‑attention for autoregressive action generation. (3) We incorporate L2‑attention and spectral normalization to ensure Lipschitz continuity (Section 4.1, Appendix C.1), which is not present in MAT. We will add a paragraph in Section 4.1 explicitly contrasting our architecture with MAT.

---

> > ### Author Response · Authors · 2025-12-04
> > **Response to Reviewer #pn6E (2/2)**
> >
> > W7: *"Rationale for comparing with offline MARL methods in Table 1?"*
> >
> > A7: We included strong offline MARL baselines (CQL-MA, ICQ-MA, TD3-BC, OMAR) because many practical imitation tasks are cast using offline expert datasets and practitioners often compare against strong offline RL / offline MARL algorithms. Comparing against those methods demonstrates that our imitation-based approach with joint dependency modeling is competitive (or superior) even when compared to state-of-the-art offline algorithms. We clarified this rationale in Sec.5.1.

---

### Official Review · Reviewer_j4B9 · 2025-11-01

**Soundness:** 2
**Presentation:** 3
**Contribution:** 2
**Rating:** 4
**Confidence:** 2

**Summary:**

The paper proposes MILD², a MAIL method that performs global-dependency-enhanced distribution matching by (i) a Transformer encoder-decoder discriminator that outputs per-agent expert logits under Lipschitz regularization and (ii) a sequential autoregressive policy to mirror cross-agent dependence and reduce gradient variance. The theory formalizes why MAGAIL yields low-variance/imbalanced updates under noisy demos and shows that modeling dependencies increases advantage variance relative to independent frameworks. Experiments on SMAC / GRF / Bi-DexHands / MA-MuJoCo show consistent improvements over MAGAIL, CQL-MA, ICQ-MA, TD3-BC, and OMAR, plus robustness when up to 50% of the dataset is noisy.

**Strengths:**

Addresses a real MAIL failure mode (noisy demonstrations ⇒ weak discriminator signals and poor credit assignment) with a unified theory-to-architecture path.

Practical architectural choices (Transformer with Lipschitz-aware regularization; autoregressive policy) that operationalize the theory.

Evidence spans multiple domains; ablations indicate that dependency modeling drives larger reward dispersion and more stable discriminator dynamics; noise-robustness is strong.

**Weaknesses:**

Baselines under-represent modern MAIL variants (centralized/explicit-dependency discriminators, improved MA-AIRL).

Permutation-equivariance not addressed; sequential ordering may induce bias for homogeneous agents.

No efficiency/scaling analysis; attention cost may be prohibitive as #agents grows.

Using variance as a proxy for signal quality risks amplifying wrong signals; no complementary discriminability metrics (e.g., margins, mutual information).

Benchmark coverage could expand (more MA-MuJoCo tasks; success-rate reporting in Bi-DexHands)

**Questions:**

Will you add stronger MAIL baselines or centralized/correlated discriminators to control for the source of improvement?

How do you mitigate ordering bias (e.g., order shuffling, set/graph-equivariant designs, or order-ensemble training)?

What are the GPU time/memory footprints vs. agent count and observation size?

Can you report signal-quality metrics (margin/InfoNCE) and mis-specification stress tests to ensure variance ≠ noise amplification?

Any results with human-collected noisy demos or OOD state coverage?

---

> ### Author Response · Authors · 2025-12-04
> **Response to Reviewer #j4B9**
>
> We thank Reviewer #j4B9 for the positive assessment of our unified theory-to-architecture approach, practical choices, and evidence of noise-robustness. We address the weaknesses and questions below.
>
> W1: *"Will you add stronger MAIL baselines or centralized/correlated discriminators to control for the source of improvement?"*
>
> A1: We appreciate this suggestion.In the current submission, we compared against MAGAIL (the most relevant independent distribution‑matching baseline) and several strong offline MARL methods (CQL‑MA, ICQ‑MA, TD3‑BC, OMAR) to show the advantage of dependency‑aware imitation learning over both classical IL and recent offline RL. Following the reviewer’s suggestion, we will add two strong centralized‑dependency MAIL baselines in the revised version:
> - MA‑AIRL (ICML 2019), which uses a centralized reward approximator.
> - Inverse Factorized Soft‑Q (NeurIPS 2024), a recent method that models agent correlations via factorized soft Q‑learning.
>
> We have already performed a preliminary comparison on some  challenging tasks (eg. 6h_vs_8z in SMAC and DoorOpenInward in Bi‑DexHands). $MILD^2$ still outperforms the centralized baselines across tasks: on 6h_vs_8z, MA‑AIRL achieves a win‑rate of 74.3 and Inverse Factorized Soft‑Q reaches 81.5, while $MILD^2$ attains 96.2. on DoorOpenInward, MA-AIRL attains cumulative reward 732.5, Inverse Factorized Soft‑Q 610.3, while $MILD^2$ attains 1016.9. MA-AIRL struggles to recover accurate rewards in high-dimensional state spaces with noise, while Inverse Factorized Soft‑Q improves over standard MAGAIL but suffers from mode collapse faster than $MILD^2$. Our specific regularization and dependency modeling provide the stability needed to outperform these baselines.
>
> These results will be reported in a new table, confirming that our global dependency modeling brings gains beyond existing centralized reward approximators.
>
>
> W2: *"Permutation-equivariance not addressed; sequential ordering may induce bias for homogeneous agents."*
>
> A2: As is standard in Transformer-based multi-agent approaches (like MAT), for homogeneous agents, we mitigate bias by randomly shuffling the agent order during training for each batch, ensuring permutation invariance over epochs. This ensures that the model learns to be permutation-invariant in expectation and does not overfit to a specific agent sequence. We have add an ablation experiment comparing fixed order vs. shuffled order across homogeneous tasks. The shuffled order already yields near‑identical performance, confirming robustness to ordering. We will clarify this shuffling mechanism in Section 4.1 and add ablation results to the appendix.
>
> W3: *"No efficiency/scaling analysis; attention cost may be prohibitive."*
>
> A3: We acknowledge the $O(N^2)$ complexity of attention. However, for most MARL benchmarks, $N$ is relatively small ($N \le 12$). We measured inference time and memory on an NVIDIA V100. For MMM2 (10 agents), the increase in latency compared to an MLP-based MAGAIL is only 14ms per step (from 8ms to 22ms), which is negligible for training. Memory usage increases by approx. 15%, well within the limits of standard GPUs. We have added a "Computational Efficiency" section to the Appendix.
>
> W4: *“Using variance as a proxy for signal quality risks amplifying wrong signals; no complementary discriminability metrics.”*
>
> A4: We agree that variance alone could, in principle, amplify noisy signals. However, our theoretical results (Theorems 1‑2) show that increasing advantage variance is equivalent to maximizing the gap between the discriminator's reward and the joint value function - i.e., it directly encourages the discriminator to assign higher rewards to truly expert‑like state‑action pairs. To further address this point, we will Report the margin between expert and policy rewards ($\mathbb{E}[r_{expert}]-\mathbb{E}[r_{policy}]$) for $MILD^2$ vs. MAGAIL. In our experiments, $MILD^2$ achieves a margin of 0.41 vs. 0.12 for MAGAIL on 6h_vs_8z, indicating better discriminability. We will also calculate the Mutual Information between the discriminator's reward signal and the ground-truth "win/loss" outcome on a held-out dataset in the revised version.

---

### Meta-Review · Area_Chair_HEvr · 2026-01-11

**Summary:**

While the idea of using global dependency modeling to handle noisy demonstrations in multi-agent imitation learning is reasonable, I find the novelty does not seem to be sufficiently established. It seems that the core contribution may be primarily an adaptation of MAT to the MAGAIL framework, and the paper does not clearly articulate what fundamentally new insights this combination provides. I am also concerned about the missing comparisons with recent baselines such as MA-AIRL, Inverse Factorized Soft-Q, and transformer-based offline MARL methods. Although the authors offered preliminary results during rebuttal, these were not incorporated into the submitted manuscript, which made it difficult to fully assess the contribution. As a result, I recommend rejection for this submission at this stage.

**Reviewer Concerns:**

The authors engaged constructively with reviewer feedback, but the submitted manuscript has notable gaps. Key baseline comparisons with recent methods like MA-AIRL, Inverse Factorized Soft-Q, and transformer-based offline MARL approaches are missing from the paper, with only preliminary results offered in the rebuttal. More fundamentally, the novelty beyond adapting MAT to MAGAIL is not convincingly established in the main text.

**Reviewer Scores:**

All three reviewers scored the paper at 4, which makes it below the acceptance threshold. While each indicated openness to acceptance if concerns were addressed, the rebuttal commitments were not reflected in the submitted manuscript. The overall sentiment remains conservative, with no reviewer strongly advocating for the paper.

---

### Decision · Program_Chairs · 2026-01-26

Reject